# WILDLY UNSUPERVISED DOMAIN ADAPTATION AND ITS POWERFUL AND EFFICIENT SOLUTION

## ABSTRACT

In *unsupervised domain adaptation* (UDA), classifiers for the *target domain* (TD) are trained with *clean* labeled data from the *source domain* (SD) and unlabeled data from TD. However, in the wild, it is hard to acquire a large amount of perfectly clean labeled data in SD given limited budget. Hence, we consider a new, more realistic and more challenging problem setting, where classifiers have to be trained with *noisy* labeled data from SD and unlabeled data from TD—we name it *wildly UDA* (WUDA). We show that WUDA ruins all UDA methods if taking no care of label noise in SD, and to this end, we propose a *Butterfly* framework, a powerful and efficient solution to WUDA. Butterfly maintains four models (e.g., deep networks) simultaneously, where two take care of all adaptations (i.e., noisy-to-clean, labeled-to-unlabeled, and SD-to-TD-distributional) and then the other two can focus on classification in TD. As a consequence, Butterfly possesses all the conceptually necessary components for solving WUDA. Experiments demonstrate that under WUDA, Butterfly significantly outperforms existing baseline methods.

## 1 INTRODUCTION

*Domain adaptation* (DA) aims to learn a discriminative classifier in the presence of a shift between training data in source domain and test data in target domain (Ben-David et al., 2010; Ganin and Lempitsky, 2015; Xiao and Guo, 2015; Zhang et al., 2015; 2013). Currently, DA can be divided into three categories: *supervised DA* (Tzeng et al., 2015), *semi-supervised DA* (Guo and Xiao, 2012) and *unsupervised DA* (UDA) (Saito et al., 2017). When the number of labeled data is few in target domain, supervised DA is also known as *few-shot DA* (Motiian et al., 2017). Since unlabeled data in target domain can be easily obtained, UDA exhibits the greatest potential in the real world (Ganin and Lempitsky, 2015; Ganin et al., 2016; Gong et al., 2012; 2016; Long et al., 2015; Saito et al., 2017; 2018).

UDA methods train with clean labeled data in source domain (i.e., clean source data) and unlabeled data in target domain (i.e., unlabeled target data) to obtain classifiers for the *target domain* (TD), which mainly consist of three orthogonal techniques: *integral probability metrics* (IPM) (Ghifary et al., 2017; Gong et al., 2016; Gretton et al., 2012; Lee and Raginsky, 2018; Long et al., 2015), *adversarial training* (Ganin et al., 2016; Gong et al., 2018; Hoffman et al., 2018; Li et al., 2018; Saito et al., 2018; Tzeng et al., 2017) and *pseudo labeling* (Saito et al., 2017). Compared to IPM- and adversarial-training-based methods, the pseudo-labeling-based method (i.e., *asymmetric tri-training domain adaptation* (ATDA) (Saito et al., 2017)) can construct a high-quality target-specific representation, providing a better classification performance.

However, in the wild, the data volume of source domain tends to be large. To avoid the expensive labeling cost, labeled data in source domain normally come from amateur annotators or the Internet (Lee et al., 2018; Schroff et al., 2011; Tommasi and Tuytelaars, 2014). This brings us a new, more realistic and more challenging problem, *wildy unsupervised domain adaptation* (abbreviated as WUDA, Figure 1). This adaptation aims to

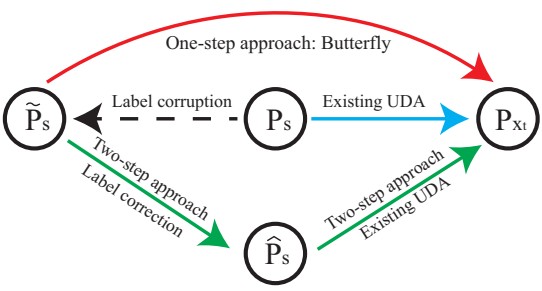

The blue line denotes that UDA transfers knowledge from clean source data ($P_s$) to unlabeled target data ($P_{x_t}$). However, perfectly clean data is hard to acquire. This brings *wildly unsupervised domain adaptation* (WUDA), namely transferring knowledge from noisy source data ($\widetilde{P}_s$) to unlabeled target data ($P_{x_t}$). Note that label corruption process (black dash line) is unknown in practice. To handle WUDA, a compromise solution is a two-step approach (green line), which sequentially combines label-noise algorithms ($\widetilde{P}_s \to \hat{P}_s$, label correction) and existing UDA ($\hat{P}_s \to P_{x_t}$). This paper proposes a robust one-step approach called Butterfly (red line, $\widetilde{P}_s \to P_{x_t}$ directly), which eliminates noise effects from $\tilde{P}_s$.

Figure 1: Wildly unsupervised domain adaptation (WUDA).

transfer knowledge from noisy labeled data in source domain ($\widetilde{P}_s$, i.e., noisy source data) to unlabeled target data ($P_{x_t}$). Unfortunately, existing UDA methods share an implicit assumption that *there are no noisy source data*. Namely, these methods focus on transferring knowledge from clean source data ($P_s$) to unlabeled target data ($P_{x_t}$). Therefore, these methods cannot well handle the WUDA.

To validate this fact, we empirically reveal the deficiency of existing UDA methods (Figure 2). To improve these methods, a straightforward solution is a two-step approach. In Figure 1, we can first use label-noise algorithms to train a model on noisy source data, then leverage this trained model to assign pseudo labels for noisy source data. Via UDA methods, we can transfer knowledge from pseudo-labeled source data ($\hat{P}_s$) to unlabeled target data ($P_{x_t}$). Nonetheless, pseudo-labeled source data are still noisy, and such two-step approach may not eliminate noise effects.

To circumvent the issue of two-step approach, we present a robust one-step approach called *Butterfly*. In high level, Butterfly directly transfers knowledge from $\widetilde{P}_s$ to $P_{x_t}$, and uses the transferred knowledge to construct target-specific representations. In low level, Butterfly maintains four networks dividing two branches (Figure 3): Two networks in Branch-I are jointly trained on noisy source data and pseudo-labeled target data (data in *mixture domain* (MD)); while two networks in Branch-II are trained on pseudo-labeled target data.

The reason why Butterfly can be robust takes root in the *dual-checking principle* (DCP): Butterfly checks high-correctness data out, from not only the data in MD but also the pseudo-labeled target data. After cross-propagating these high-correctness data, Butterfly can obtain high-quality *domain-invariant representations* (DIR) and *target-specific representations* (TSR) simultaneously in an iterative manner. If we only check data in MD (i.e., single checking), the error existed in pseudo-labeled target data will accumulate, leading to low-quality DIR and TSR.

We conduct experiments on simulated WUDA tasks, including 4 *MNIST-to-SYND* tasks, 4 *SYND-to-MNIST* tasks and 24 human-sentiment tasks. Besides, we conduct experiments on 3 real-world WUDA tasks. Empirical results demonstrate that Butterfly can robustly transfer knowledge from noisy source data to unlabeled target data. Meanwhile, Butterfly performs much better than existing UDA methods when *source domain* (SD) suffers the extreme (e.g., $45\%$) noise.

## 2 WILDLY UNSUPERVISED DOMAIN ADAPTATION

In this section, we first define a new, more realistic and more challenging setting called *wildly unsupervised domain adaptation (WUDA)*, and explain the nature of WUDA. Then, we empirically show that representative UDA methods cannot handle WUDA well, which motivates us to propose Butterfly (see Section 3).

**Definition 1** (Wildly Unsupervised Domain Adaptation). *Let $X_t$ be a multivariate random variable defined on the space $\mathcal{X}$ with respective a probability density $p_{x_t}$, $(X_s, Y_s)$ be a multivariate random variable defined*

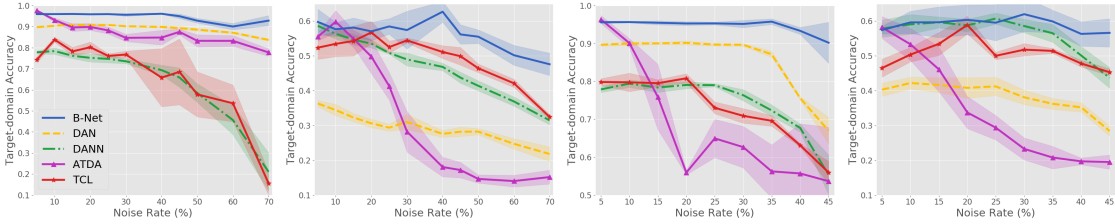

(a) Symmetry-flip noise: $S{\rightarrow}M$ (left), $M{\rightarrow}S$ (right)     (b) Pair-flip noise: $S{\rightarrow}M$ (left), $M{\rightarrow}S$ (right)

Figure 2: WUDA ruins representative UDA methods. Representative UDA methods includes *deep adaptation networks* (DAN, a IPM based method (Long et al., 2015)), *domain-adversarial neural network* (DANN, a adversarial training based method (Ganin et al., 2016)), *asymmetric tri-training domain adaptation* (ATDA, a pseudo-label based method (Saito et al., 2017)) and *transferable curriculum learning* (TCL, a robust UDA method (Shu et al., 2019)). B-Net is our proposed WUDA method. We report target-domain accuracy of all methods when the noise rate of source domain changes (a) from 5% to 70% (symmetry-flip noise) and (b) from 5% to 45% (pair-flip noise). Clearly, when the noise rate of source domain increases, target-domain accuracy of representative UDA methods drops quickly while that of B-Net keeps stable consistently.

*on the space $\mathcal{X} \times \mathbb{C}$ with respective a probability density $\tilde{p}_s$, where $\mathbb{C} = \{1, \ldots, K\}$. Let $p_{x_s}$ be the marginal probability density of $\tilde{p}_s$. Given i.i.d. data $\tilde{D}_s = \{(x_{si}, \tilde{y}_{si})\}_{i=1}^{n_s}$ and $D_t = \{x_{ti}\}_{i=1}^{n_t}$ drawn from $\tilde{P}_s$ and $P_{x_t}$, in wildly unsupervised domain adaptation, we aim to train with $\tilde{D}_s$ and $D_t$ to accurately annotate data drawn from $P_{x_t}$, where $p_{x_s} \neq p_{x_t}$.*

**Remark 1.** In Definition 1, $\tilde{D}_s$ is noisy source data, $D_t$ is unlabeled target data, and $\tilde{P}_s$ and $P_{x_t}$ are two probability measures corresponding to densities $\tilde{p}_s(x_s, \tilde{y}_s)$ and $p_{x_t}(x_t)$.

**Nature of WUDA.** Specifically, there are five distributions involved in WUDA setting: 1) a marginal distribution on source data, i.e., $p_{x_s}$ in Definition 1; 2) a marginal distribution on target data, i.e., $p_{x_t}$ in Definition 1; 3) an incorrect conditional distribution of label given $x_s$, $q(y_s|x_s)$; 4) a correct conditional distribution of label given $x_s$, $p(y_s|x_s)$ and 5) a correct conditional distribution of label given $x_t$, $p(y_t|x_t)$.

Based on Definition 1 and Appendix A.2, noisy source data $\tilde{D}_s$ are drawn from $\tilde{p}_s(x_s, y_s) = p_{x_s}(x_s)((1 - \rho)p(y_s|x_s) + \rho q(y_s|x_s))$, where $\rho$ is the noise rate in source data. Namely, source data $\tilde{D}_s$ are mixture of correct source data from $p_{x_s}(x_s)p(y_s|x_s)$ and incorrect data from $p_{x_s}(x_s)q(y_s|x_s)$. Target data $D_t$ are drawn from $p_{x_t}$. In WUDA setting, we aim to train a classifier with $\tilde{D}_s$ and $D_t$. This classifier is expected to accurately annotate data from $p_{x_t}$, i.e., to accurately simulate distribution 5).

**WUDA ruins representative UDA methods.** We take a simple example to illustrate why WUDA ruins representative UDA methods. We corrupt source data using symmetry flipping (Patrini et al., 2017) and pair flipping (Han et al., 2018) (Appendix B). Namely, the corrupted source data ($\tilde{D}_s$ in Definition 1) are drawn from $\tilde{P}_s$ whose density is $\tilde{p}_s(x_s, y_s) = p_{x_s}(x_s)((1 - \rho)p(y_s|x_s) + \rho q(y_s|x_s))$. We draw the target data $D_t$ from $P_{x_t}$ whose density is $p_{x_t}$. To instantiate source and target data, we leverage *MNIST* and *SYND* (Figure 6 in Appendix C), respectively.

Thus, we first construct two WUDA tasks with symmetry-flip noise: corrupted *SYND→MNIST* ($S{\rightarrow}M$) and corrupted *MNIST→SYND* ($M{\rightarrow}S$). In Figure 2 (a), we report target-domain accuracy of representative UDA methods, when the noise rate $\rho$ of SD changes from 5% to 70%. It is clear that target-domain accuracy of these representative UDA methods drops quickly when $\rho$ increases. This means that WUDA ruins representative UDA methods. Then, we construct another two WUDA tasks with pair-flip noise. In Figure 2 (b), we report target-domain accuracy, when the noise rate $\rho$ of SD changes from 5% to 45%. Again, WUDA still ruins representative UDA methods. Note that pair-flip noise is much harder than symmetry-flip noise, and its noise rate cannot be over 50% in practice (Han et al., 2018).

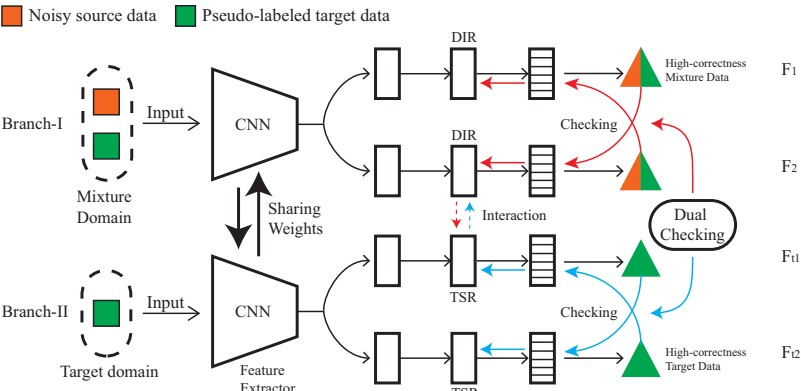

Figure 3: Butterfly Framework. Two networks ($F_1$ and $F_2$) in Branch-I are jointly trained on noisy source data and pseudo-labeled target data (mixture domain). Two networks in Branch-II ($F_{t1}$ and $F_{t2}$) are trained on pseudo-labeled target data. By using dual-checking principle, Butterfly checks high-correctness data out from both mixture and pseudo-labeled target data. After cross-propagating checked data, Butterfly can obtain high-quality *domain-invariant representations* (DIR) and *target-specific representations* (TSR) simultaneously in an iterative manner. Note that the interaction between DIR and TSR happens via the shared CNN. Besides, in the first training epoch, since we do not have any pseudo-labeled target data, we need to use noisy source data as the pseudo-labeled target data, which follows Saito et al. (2017).

However, the proposed Butterfly network (abbreviated as B-Net, Figure 3) performs robustly when $\rho$ increases (blue lines in Figure 2). In following sections, we will introduce Butterfly framework, and explain why Butterfly achieves better target-domain accuracy consistently.

## 3 BUTTERFLY: TOWARDS ROBUST ONE-STEP APPROACH

To realize a robust WUDA approach, we propose a Butterfly framework (a one-step approach, Algorithm 1), which trains four networks dividing into two branches (Figure 3). By using DCP, Branch-I checks which data is correct in MD; while Branch-II checks which pseudo-labeled target data is correct. To ensure these checked data highly-correct, we apply the small-loss trick based on memorization effects of deep learning (Arpit et al., 2017). After cross-propagating these checked data (Bengio, 2014), Butterfly can obtain high-quality DIR and TSR simultaneously in an iterative manner.

**Loss function in Butterfly.** Four networks trained by Butterfly share the same loss function but with different inputs.

$$\mathcal{L}(\theta, u; F, D) = \frac{1}{\sum_{i=1}^{n} u_i} \sum_{i=1}^{n} u_i \ell(F(x_i), \check{y}_i), \tag{1}$$

where $n$ is the batch size, and $F$ represents a network (e.g., $F_1$, $F_2$, $F_{t1}$ and $F_{t2}$). $D = \{(x_i, \check{y}_i)\}_{i=1}^{n}$ is a mini-batch for training a network, where $\{x_i, \check{y}_i\}_{i=1}^{n}$ could be data in MD or TD (Figure 3), and $\theta$ represents parameters of $F$ and $u = [u_1, ..., u_n]^T$ is an $n$-by-1 vector whose elements equal 0 or 1. $\ell(\cdot, \cdot)$ is the cross-entropy loss. For two networks in Branch-I, following Saito et al. (2017), we also add a regularizer $|\theta_{f11}^T \theta_{f21}|$ in their loss functions, where $\theta_{f11}$ and $\theta_{f21}$ are weights of the first fully-connect layer of $F_1$ and $F_2$. With this regularizer, $F_1$ and $F_2$ will learn from different features.

**Nature of the loss $\mathcal{L}$.** In loss function $\mathcal{L}$, we have $n$ instances: $(x_i, \check{y}_i)$, where $i = 1, 2, \ldots, n$. For the $i^{th}$ instance, we will compute its cross-entropy loss (i.e., $\ell(F(x_i), \check{y}_i)$), and we will denote this instance as "selected" if $u_i = 1$. Thus, the nature of $\mathcal{L}$ is actually the average value of cross-entropy loss of these

"selected" instances. Note that, we need to set a constrain to prevent $\sum_{i=1}^{n} u_i = 0$ in $\mathcal{L}$, which means that we need to select at least one instance to compute $\mathcal{L}$.

**Training procedure of Butterfly.** For two networks in each branch, they will first check high-correctness data out and then cross update their parameters using these data.

Based on loss function defined in Eq. (1), entire training procedures of Butterfly are shown in Algorithm 1. First, we initialize training data for two branches ($\tilde{D}$ for Branch-I and $\tilde{D}_t^l$ for Branch-II), four networks ($F_1, F_2, F_{t1}$ and $F_{t2}$) and the number of pseudo labels (line 2). In the first epoch ($T = 1$), following Saito et al. (2017), $\tilde{D}_t^l$ is the same with $\tilde{D}_s$ (i.e., we use noisy source data as pseudo-labeled target data) since we cannot annotate pseudo labels for target data when $T = 1$. After mini-batch $\check{D}$ is fetched from $\tilde{D}$ (line 4), $F_1$ and $F_2$ check high-correctness data out and update their parameters (lines 5) using Algorithm 2. Using similar procedures, $F_{t1}$ and $F_{t2}$ also update their parameters using Algorithm 2 (lines 6-7).

In each epoch, after $N_{max}$ mini-batch updating, we randomly select $n_t^l$ unlabeled target data and assign them pseudo labels using $F_1$ and $F_2$ (lines 8). Following Saito et al. (2017), the Labeling function in Algorithm 1 (line 8) assigns pseudo labels for unlabeled target data, when predictions of $F_1$ and $F_2$ agree and at least one of them is confident about their predictions (probability above 0.9 or 0.95). Using this function, we can obtain the pseudo-labeled target data $\tilde{D}_t^l$ for training Branch-II in the next epoch. Then, we merge $\tilde{D}_t^l$ and $\tilde{D}_s$ to be $\tilde{D}$ for training Branch-I in the next epoch (line 9). Finally, we update $n_t^l$, $R(T)$ and $R_t(T)$ in lines 10-11 according to Saito et al. (2017) and Han et al. (2018). Note that $R(T)$ is a piecewise-defined linear function. Namely, when $T \geq T_k$, $R(T) = 1 - \tau$; when $T \leq T_k$, $R(T) = 1 - T/T_k \times \tau$.

In Algorithm 1, we use $\tau$ to represent the noise rate (i.e., the ratio of data with incorrect labels) in MD and use $\tau_t$ to represent the noise rate in TD. However, in WUDA, we cannot obtain the ground-truth $\tau$ and $\tau_t$. Thus, we regard $\tau$ and $\tau_t$ as hyper-parameters. Note that $\tau$ and $\tau_t$ are robustly set to $0.4$ and $0.05$ in experiments.

**Checking process in Butterfly.** In Algorithm 2, we first obtain four inputs: 1) networks $F_1$ and $F_2$, and 2) a mini-batch $D$, and 3) learning rate $\eta$ and 4) remember rate $\alpha$ (line 1). Then, we will obtain the best $\boldsymbol{u}_1$ by solving a minimization problem (line 2). $\mathcal{L}$ represents the loss function defined in Eq. (1). $\theta_1$ represents the parameters of the network $F_1$. Similarly we will obtain the best $\boldsymbol{u}_2$ (line 3). $\theta_2$ represents the parameters of the network $F_2$. Next, $\theta_1$ and $\theta_2$ are updated using gradient descent, where the gradients are computed using a given optimizer (lines 4-5). Finally, we substitute the updated $\theta_1$ into $F_1$ and the updated $\theta_2$ into $F_2$ and output $F_1$ and $F_2$ (line 6).

**Solution to minimization problems in Algorithm 2.** In line 2 or 3 in Algorithm 2, we need to solve a minimization problem: $\min_{\boldsymbol{u}':\boldsymbol{1}\boldsymbol{u}'>\alpha|D|} \mathcal{L}(\theta, \boldsymbol{u}'; F, D)$ and return the best $\boldsymbol{u}'$ as $\boldsymbol{u}$ ($\boldsymbol{u}_1$ in line 2 and $\boldsymbol{u}_2$ in line 3). In this paragraph, we will show how to quickly solve this problem using a sorting algorithm. Recall the nature of the loss $\mathcal{L}$, we know $\mathcal{L}$ is the average value of cross-entropy losses of "selected" instances, and $\boldsymbol{1}\boldsymbol{u}'$ is the number of these "selected" instances. Thus this minimization problem is equivalent to "given a fixed $F$ ($F_1$ or $F_2$) and $n$ instances in $D$, how to select at least $k$ instances such that $\mathcal{L}$ is minimized", where $k = \lceil \alpha|D| \rceil$. To solve this problem, we first use a sorting algorithm (top_k function in TensorFlow) to sort these $n$ instances according to their cross-entropy losses $\ell(F_1(x_i), \check{y}_i)$. Then, we select $k$ instance with the smallest cross-entropy losses. Finally, let $u_i$ of these $k$ instances be 1 and $u_i$ of the other instances be 0, and we can get the best $\boldsymbol{u} = [u_1, \ldots, u_n]$. The average value of cross-entropy losses of these $k$ instances is the minimized value of $\mathcal{L}(\theta, \boldsymbol{u}'; F, D)$ under the constrain $\boldsymbol{1}\boldsymbol{u}' > \alpha|D|$.

## 4 BUTTERFLY VS. TWO-STEP APPROACH

This section analyzes why Butterfly is better than two-step approach using theoretical results in Appendix D. Practitioner may safely skip it. Following Ben-David et al. (2010), we derive an upper bound of target-domain risk for WUDA. Compared to existing UDA bounds, the WUDA bound has two extra terms (see Eq. (6)): $\Delta_s$ (noise effect from source data), and $\Delta_t$ (noise effects from pseudo labels of target data). We will use $\Delta_s$ and $\Delta_t$ to show why Butterfly (a one-step approach) can eliminate noise effects while two-step methods cannot.

---

**Algorithm 1** Butterfly Framework: quadruple training for WUDA problem

---

1: **Input** $\tilde{D}_s$, $D_t$, learning rate $\eta$, fixed $\tau$, fixed $\tau_t$, epoch $T_k$ and $T_{max}$, iteration $N_{max}$, # of pseudo-labeled target data $n_{init}$, max of $n_{init}$ $n_{t,max}^l$;

2: **Initial** $F_1$, $F_2$, $F_{t1}$, $F_{t2}$, $\tilde{D}_t^l = \tilde{D}_s$, $\tilde{D} = \tilde{D}_s$, $n_t^l = n_{init}$;

**for** $T = 1, 2, \ldots, T_{max}$ **do**

   3: **Shuffle** training set $\tilde{D}$;      // Noisy dataset

   **for** $N = 1, \ldots, N_{max}$ **do**

      4: **Fetch** mini-batch $\check{D}$ from $\tilde{D}$;

      5: **Update** Branch-I: $F_1, F_2 =$ Checking($F_1, F_2, \check{D}, \eta, R(T)$);    // Check data in MD using Algorithm 2

      6: **Fetch** mini-batch $\check{D}_t$ from $\tilde{D}_t^l$;

      7: **Update** Branch-II: $F_{t1}, F_{t2} =$ Checking($F_{t1}, F_{t2}, \check{D}_t, \eta, R_t(T)$);    // Check data in TD using Algorithm 2

   **end**

   8: **Obtain** $\tilde{D}_t^l =$ Labelling($F_1, F_2, D_t, n_t^l$);    // Label $D_t$, following Saito et al. (2017)

   9: **Obtain** $\tilde{D} = \tilde{D}_s \cup \tilde{D}_t^l$;    // Update MD

   10: **Update** $n_t^l = \min\{T/20 * n_t, n_{t,max}^l\}$;

   11: **Update** $R(T) = 1 - \min\{\frac{T}{T_k}\tau, \tau\}$, $R_t(T) = 1 - \min\{\frac{T}{T_k}\tau_t, \tau_t\}$;

**end**

12: **Output** $F_{t1}$ and $F_{t2}$

---

**Algorithm 2** Checking($F_1, F_2, D, \eta, \alpha$)

---

1: **Input** networks $F_1$, $F_2$, mini-batch $D$, learning rate $\eta$, remember rate $\alpha$;

2: **Obtain** $\boldsymbol{u}_1 = \arg\min_{\boldsymbol{u}_1': \mathbf{1}\boldsymbol{u}_1' > \alpha|D|} \mathcal{L}(\theta_1, \boldsymbol{u}_1'; F_1, D)$;    // Check high-correctness data

3: **Obtain** $\boldsymbol{u}_2 = \arg\min_{\boldsymbol{u}_2': \mathbf{1}\boldsymbol{u}_2' > \alpha|D|} \mathcal{L}(\theta_2, \boldsymbol{u}_2'; F_2, D)$;    // Check high-correctness data

4: **Update** $\theta_1 = \theta_1 - \eta\nabla\mathcal{L}(\theta_1, \boldsymbol{u}_2; F_1, D)$;    // Update $\theta_1$

5: **Update** $\theta_2 = \theta_2 - \eta\nabla\mathcal{L}(\theta_2, \boldsymbol{u}_1; F_2, D)$;    // Update $\theta_2$

6: **Output** $F_1$ and $F_2$

---

**Two-step approach (a compromise solution).** To reduce noise effects, a straightforward solution is two-step approach. In the first step, we can train a classifier with noisy source data using Co-teaching (Han et al., 2018) and use this classifier to annotate pseudo labels for source data. In the second step, we use ATDA (Saito et al., 2017) to train a target-domain classifier with pseudo-label-source data and target data.

However, two-step approach may not reduce noise effects $\Delta_s$ (i.e., not alleviating noise effects from source data). In two-step approach, after using Co-teaching, $\Delta_s$ will become pseudo-label-source effects $\Delta_s'$ (see Eq. (7)). The first part of $\Delta_s'$ may be less than that of $\Delta_s$ due to Co-teaching, but the second term of $\Delta_s'$ may be higher than that of $\Delta_s$ since Co-teaching does not consider to minimize it. Thus, it is hard to say whether $\Delta_s' < \Delta_s$. This means that, the two-step approach may not really reduce noise effects $\Delta_s$. Besides, two-step approach does not take care of eliminating $\Delta_t$ explicitly. Based on above analysis, we can find that a two-step approach cannot eliminate $\Delta_s$ and $\Delta_t$.

**One-step approach (Butterfly).** To eliminate noise effects $\Delta = \Delta_s + \Delta_t$, Butterfly aims to select correct data simultaneously from noisy source data and pseudo-labeled target data (see Section 3). Let $\rho_{01}^s$ be the probability that incorrect data is selected from noisy source data, and $\rho_{01}^t$ be the probability that incorrect data is selected from pseudo-labeled target data. Theorem 3 shows that $\Delta \to 0$ if $\rho_{01}^s \to 0$ and $\rho_{01}^t \to 0$. Since Butterfly can select correct data with a high probability (i.e., $\rho_{01}^s \to 0$ and $\rho_{01}^t \to 0$), noise effects will be eliminated ($\Delta \to 0$).

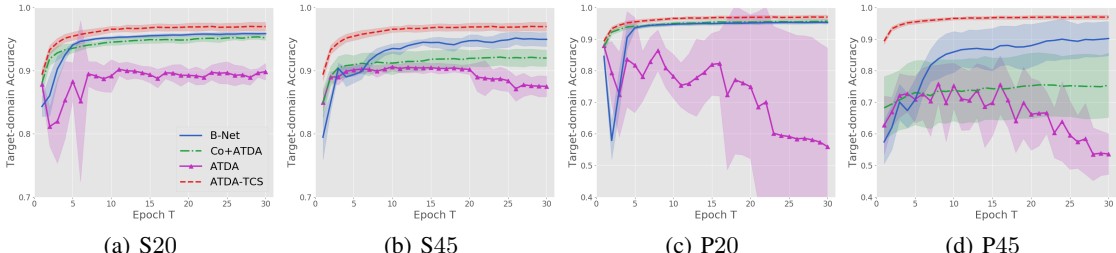

Figure 4: Target-domain accuracy vs. number of epochs on four *SYND→MNIST* WUDA tasks.

## 5 EXPERIMENTS

**Simulated WUDA tasks.** We verify the effectiveness of our approach on three benchmark datasets (vision and text), including *MNIST*, *SYN-DIGITS (SYND)* and *human-sentiment* analysis (i.e., *Amazon products reviews* on *book*, *dvd*, *electronics* and *kitchen*). They are used to construct 14 basic tasks: *MNIST→SYND* (*M→S*), *SYND→MNIST* (*S→M*), *book→dvd* (*B→D*), *book→electronics* (*B→E*), . . . , and *kitchen → electronics* (*K→E*). These tasks are often used for evaluation of UDA methods (Ganin et al., 2016; Saito et al., 2017; 2018). Since all source datasets are clean, we corrupt source data using symmetry flipping (Patrini et al., 2017) and pair flipping (Han et al., 2018) (Appendix B) with noise rate $\rho$ chosen from $\{0.2, 0.45\}$. So, for each basic task, we have four kinds of noisy source data: *Pair*-45% (P45), *Pair*-20% (P20), *Symmetry*-45% (S45), *Symmetry*-20% (S20). Namely, we evaluate the performance of each method using 32 simulated WUDA tasks: 8 digit tasks and 24 human-sentiment tasks. Note that the human-sentiment task is a binary classification problem, so pair flipping is equal to symmetry flipping. Thus, we only have 24 human-sentiment tasks. Results on human-sentiment tasks are reported in Appendix E.

**Real-world WUDA tasks.** We also verify the efficacy of our approach on "cross-dataset benchmark" including *Bing*, *Caltech256*, *Imagenet* and *SUN* (Tommasi and Tuytelaars, 2014). In this benchmark, *Bing*, *Caltech256*, *Imagenet* and *SUN* contain common 40 classes. Since *Bing* dataset was formed by collecting images retrieved by Bing image search, it contains rich noisy data, with presence of multiple objects in the same image and caricaturization (Tommasi and Tuytelaars, 2014). We use *Bing* as noisy source data, and *Caltech256*, *Imagenet* and *SUN* as unlabeled target data, which can form three real-world WUDA tasks. Besides, we also test our method on two common UDA tasks ( *MNIST→SVHN* and *SVHN→MNIST*) to help compare with other UDA-based papers in literature.

**Baselines.** We realize Butterfly using four networks (abbreviated as B-Net) and compare B-Net with following baselines: 1) ATDA: representative pseudo label based UDA method (Saito et al., 2017); 2) *deep adaptation networks* (DAN): representative IPM based UDA method (Long et al., 2015); 3) DANN: representative adversiral training based UDA method (Ganin et al., 2016); 4) TCL: an existing robust UDA method; 5) Co teaching+ATDA (Co+ATDA): a two-step method (see Section 4); 6) Co teaching+TCL (Co+TCL): a two-step method. Implementation details are demonstrated in Appendix F.

**Results on simulated WUDA (including 8 tasks).** Table 1 reports the accuracy on the unlabled target data (i.e., target-domain accuracy) in 8 tasks. As can be seen, average target-domain accuracy of B-Net is higher than those of all baselines. On S20 case (the easiest case), most methods work well. ATDA has a satisfactory performance although it does not consider the noise effects explicitly. Then, when facing harder cases (i.e., P20 and P45), ATDA fails to transfer useful knowledge from noisy source data to unlabeled target data. When facing the hardest cases (i.e., *M→S* with P45 and S45), DANN has higher accuracy than DAN and ATDA have. However, when facing the easiest cases (i.e., *S→M* with P20 and S20), the performance of DANN is worse than that of DAN and ATDA. Although two-step method Co+ATDA (or Co+TCL) outperforms ATDA (or TCL) in all 8 tasks, it cannot beat one-step method: B-Net in terms of average target-domain accuracy. This result is an evidence for the claim in Section 4. In the task *S→M* with P20, Co+ATDA outperforms all methods (slightly higher than B-Net), since pseudo-labeled source data are almost correct.

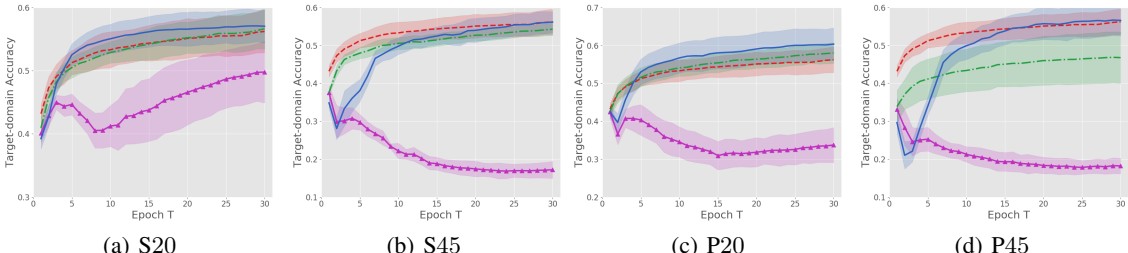

Figure 5: Target-domain accuracy vs. number of epochs on four *MNIST→SYND* WUDA tasks.

Table 1: Target-domain accuracy on 8 digit WUDA tasks (*SYND↔MNIST*). Bold value represents the highest accuracy in each row.

| Tasks | Type | DAN | DANN | ATDA | TCL | Co+TCL | Co+ATDA | B-Net |
|-------|------|-----|------|------|-----|--------|---------|-------|
| | P20 | 90.17% | 79.06% | 55.95% | 80.81% | 88.56% | **95.37%** | 95.29% |
| | P45 | 67.00% | 55.34% | 53.66% | 55.97% | 73.27% | 75.43% | **90.21%** |
| S→M | S20 | 90.74% | 75.19% | 89.87% | 80.23% | 85.88% | 95.22% | **95.88%** |
| | S45 | 89.31% | 65.87% | 87.53% | 68.54% | 75.69% | 92.03% | **94.97%** |
| | P20 | 40.82% | 58.78% | 33.74% | 58.88% | 59.08% | 58.02% | **60.36%** |
| | P45 | 28.41% | 43.70% | 19.50% | 45.31% | 47.15% | 46.80% | **56.62%** |
| M→S | S20 | 30.62% | 53.52% | 49.80% | 56.74% | 56.91% | 56.64% | **57.05%** |
| | S45 | 28.21% | 43.76% | 17.20% | 49.91% | 51.22% | 54.29% | **56.18%** |
| Average | | 58.16% | 58.01% | 50.91% | 62.05% | 67.22% | 71.73% | **75.82%** |

Figures 4 and 5 show the target-domain accuracy vs. number of epochs among ATDA, Co+ATDA and B-Net. Besides, we show the accuracy of ATDA trained with clean source data (ATDA-TCS) as a reference point. When accuracy of one method is close to that of ATDA-TCS (red dash line), this method successfully eliminates noise effects. From our observations, it is clear that B-Net is very close to ATDA-TCS in 7 out of 8 tasks (except for *S→M* task with P45, Figure 4-(d)), which is an evidence that Butterfly can eliminate noise effects. Since P45 case is the hardest one, it is reasonable that B-Net cannot perfectly eliminate noise effects. An interesting phenomenon is that, B-Net outperforms ATDA-TCS in 2 *M→S* tasks (Figure 5-(a), (c)). This means that B-Net transfers more useful knowledge (from noisy source data to unlabeled target data) even than ATDA-TCS (from clean source data to unlabeled target data).

**Results on real-world WUDA (including 3 tasks).**    Table 2 reports the target-domain accuracy for 3 tasks. B-Net enjoys the best performance on all tasks. It should be noted that, in both *Bing→Caltech256* and *Bing→ImageNet* tasks, ATDA is slightly worse than B-Net. However, in *Bing→SUN* task, ATDA is much worse than B-Net. The reason is that the DIR between *Bing* and *SUN* are more affected by noisy source data. This phenomenon is also observed when comparing DANN and TCL. Compared to Co+ATDA, ATDA is slightly better than Co+ATDA. This abnormal phenomenon can be explained using $\Delta$ (see Section 4), after using Co-teaching to assign pseudo labels for noisy source data, the second term in $\Delta_s$ may increase, which results in that $\Delta$ increases, i.e., noise effects actually increase. This phenomenon is an evidence that a two-step method may not really reduce noise effects.

**Results on two UDA tasks.**    Table 3 reports the target-domain accuracy for 2 common UDA tasks. The target-domain accuracy of DAN, DANN and ATDA are results reported by Saito et al. (2017). Although the target-domain accuracy of B-Net is not always higher than that of ATDA, B-Net still has satisfactory performance on UDA problem. Note that, since B-Net is a WUDA method, it is reasonable that B-Net is not the state-of-the-art UDA method.

**Ablation study.**    Finally, we conduct thorough experiments to show the contribution of individual components in B-Net. We report average target-domain accuracy on 32 simulated WUDA tasks (8 digit and 24

Table 2: Target-domain accuracy on 3 real-world WUDA tasks. The source domain is the *Bing* dataset that contains noisy information from the Internet. Bold value represents the highest accuracy in each row.

| Target | DAN | DANN | ATDA | TCL | Co+TCL | Co+ATDA | B-Net |
|---|---|---|---|---|---|---|---|
| *Caltech256* | 77.83% | 78.00% | 80.84% | 79.35% | 79.27% | 79.89% | **81.71%** |
| *Imagenet* | 70.29% | 72.16% | 74.89% | 72.53% | 72.33% | 74.73% | **75.00%** |
| *SUN* | 24.56% | 26.80% | 26.26% | 28.80% | 29.15% | 26.31% | **30.54%** |
| Average | 57.56% | 58.99% | 60.66% | 60.23% | 60.25% | 60.31% | **62.42%** |

Table 3: Target-domain accuracy on 2 UDA tasks. Bold value represents the highest accuracy in each row.

| Tasks | DAN | DANN | ATDA | B-Net |
|---|---|---|---|---|
| *MNIST→SVHN* | - | 35.70% | **52.80%** | 52.64% |
| *SVHN→MNIST* | 71.10% | 71.10% | 85.00% | **85.82%** |

Table 4: Results of ablation study. Average target-domain accuracy on 8 simulated digit WUDA tasks (*Digit*), 24 simulated human-sentiment WUDA tasks (*Sentiment*) and 3 real-world WUDA tasks (*Real-world*). Bold value represents the highest accuracy in each row.

| Datasets | B w/o C | DCP-D | DCP-M | B-Net-S | B-Net-T | B-Net-ST | B-Net-M | B-Net |
|---|---|---|---|---|---|---|---|---|
| *Digit* | 74.52% | 59.19% | 70.85% | 71.93% | 52.00% | 72.27% | 73.89% | **75.82%** |
| *Sentiment* | 63.57% | 61.37% | 63.39% | 61.49% | 61.12% | 61.73% | 62.21% | **63.77%** |
| *Real-world* | 62.27% | 59.82% | 62.34% | 61.91% | 60.87% | 62.24% | 62.17% | **62.42%** |

human-sentiment WUDA tasks) and 3 real-world WUDA tasks. We consider following baselines: 1) B w/o C: train B̲-Net by Algorithm 1, without adding $|\theta_{f11}^T \theta_{f21}|$ into the loss function of B-Net. 2) DCP-D: realize D̲C̲P̲ via D̲ecoupling (Malach and Shalev-Shwartz, 2017) to check data in MD and TD. 3) DCP-M: realize D̲C̲P̲ via M̲entorNet (Jiang et al., 2018) to check data in MD and TD. 4) B-Net-S: train B̲-Net where the check is turned on for S̲ource data in MD. 5) B-Net-T: train B̲-Net where the check is turned on for T̲arget data in TD. 6) B-Net-ST: train B̲-Net where the checks are turned on for S̲ource data in MD and T̲arget data in TD. 7) B-Net-M: train B̲-Net where the check is turned on for all data in M̲D̲. Note that in the full B-Net, the checks are turned on for all data in MD and Target data in TD.

Comparing B-Net with B w/o C reveals whether the constraint $|\theta_{f11}^T \theta_{f21}|$ takes effects. Comparing B-Net with DCP-D and DCP-M shows whether realizing DCP via Co-teaching is the optimal way. Comparing B-Net with B-Net-S, B-Net-T, B-Net-ST and B-Net-M reveals whether DCP is necessary. Table 4 reports average target-domain accuracy of above baselines and B-Net. As can be seen, 1) B-Net benefits from adding the constraint to the loss function $\mathcal{L}$; 2) realizing DCP by Co-teaching is better than using Decoupling or MentorNet; and 3) DCP is necessary since accuracy of B-Net is higher than those of B-Net-S, B-Net-T, B-Net-ST and B-Net-M.

## 6 CONCLUSIONS

This paper opens a new problem called *wildly unsupervised domain adaptation* (WUDA). However, existing UDA methods cannot handle WUDA well. To address this problem, we propose a robust one-step approach called *Butterfly*. Butterfly maintains four deep networks simultaneously: Two take care of all adaptations; while the other two can focus on classification in target domain. We compare Butterfly with existing UDA methods on 32 simulated and 3 real-world WUDA tasks. Empirical results demonstrate that Butterfly can robustly transfer knowledge from noisy source data to unlabeled target data. In future, we will extend our Butterfly framework to address open-set UDA when source domain contains noisy data.

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

# A  REVIEW OF GENERATION OF NOISY LABELS

This section presents a review on two label-noise generation processes.

## A.1  TRANSITION MATRIX

We assume that there is a clean multivariate random variable $(X_s, Y_s)$ defined on $\mathcal{X} \times \mathcal{Y}$ with a probability density $p_s(x_s, y_s)$, where $\mathcal{Y} = \{1, ..., K\}$ is a label set with $K$ labels. However, samples of $(X_s, Y_s)$ cannot be directly obtained and we only can observe noisy source data from the multivariate random variable $(X_s, \tilde{Y}_s)$ defined on $\mathcal{X} \times \mathcal{Y}$ with a probability density $\tilde{p}_s(x_s, \tilde{y}_s)$. $\tilde{p}_s(x_s, \tilde{y}_s)$ is generated by a transition probability $\Pr(\tilde{Y}_s = j | Y_s = i)$, i.e., the flip rate from a clean label $i$ to a noisy label $j$. When we generate $\tilde{p}_s(x_s, \tilde{y}_s)$ using $Q$, we often assume that $\sum_{y_s=1}^{K} p_s(x_s, y_s) = \sum_{\tilde{y}_s=1}^{K} \tilde{p}_s(x_s, \tilde{y}_s)$, i.e., the class conditional noise (Liu and Tao, 2016). All these transition probabilities are summarized into a transition matrix $Q$, where $Q_{ij} = \Pr(\tilde{Y}_s = j | Y_s = i)$.

The transition matrix $Q$ is easily estimated in certain situations (Liu and Tao, 2016). However, in more complex situations, such as clothing1M dataset (Xiao et al., 2015), noisy source data is directly generated by selecting data from a pool, which mixes correct data (data with correct labels) and incorrect data (data with incorrect labels). Namely, how the correct label $i$ is corrupted to $j$ ($i \neq j$) is unclear.

## A.2  SAMPLE SELECTION

Formally, there is a multivariate random variable $(X_s, Y_s, V_s)$ defined on $\mathcal{X} \times \mathcal{Y} \times \mathcal{V}$ with a probability density $p_s^{\text{po}}(x_s, y_s, v_s)$, where $\mathcal{V} = \{0, 1\}$ and $V_s = 1$ means "correct" and $V_s = 0$ means "incorrect". Nonetheless, samples from $(X_s, Y_s, V_s)$ cannot be obtained and we can only observe $(X_s, \tilde{Y}_s)$ from a distribution with the following density.

$$\tilde{p}_s(x_s, \tilde{y}_s) = \sum_{v_s=0}^{1} p_{X_s, Y_s | V_s}^{\text{po}}(x_s, y_s | v_s) p_{V_s}^{\text{po}}(v_s), \tag{2}$$

where $p_{V_s}^{\text{po}}(v_s) = \int_{\mathcal{X}} \sum_{y_s=1}^{K} p_s^{\text{po}}(x_s, y_s, v_s) dx_s$. The density in Eq. (2) means that we lost the information from $V_s$. If we uniformly select samples drawn from $\tilde{p}_s(x_s, \tilde{y}_s)$, the noise rate of these samples is $p_{V_s}^{\text{po}}(0)$. It is clear that the multivariate random variable $(X_s, Y_s | V_s = 1)$ is the clean multivariate random variable $(X_s, Y_s)$ defined in Appendix A.1. Then, $q_s(x_s, y_s)$ is used to describe the density of incorrect multivariate random variable $(X_s, Y_s | V_s = 0)$. Using $p_s(x_s, y_s)$ and $q_s(x_s, y_s)$, $\tilde{p}_s(x_s, \tilde{y}_s)$ can be expressed by the following equation.

$$\tilde{p}_s(x_s, \tilde{y}_s) = (1 - \rho) p_s(x_s, y_s) + \rho q_s(x_s, y_s), \tag{3}$$

where $\rho = p_{V_s}^{\text{po}}(0)$. Here, we do not assume $\sum_{y_s=1}^{K} p_s(x_s, y_s) = \sum_{y_s=1}^{K} q_s(x_s, y_s)$. To reduce noise effects from incorrect data, scholars aim to recover the information of $V_s$, i.e., to select correct data from data drawn from $\tilde{p}_s(x_s, \tilde{y}_s)$ (Han et al., 2018; Jiang et al., 2018; Malach and Shalev-Shwartz, 2017).

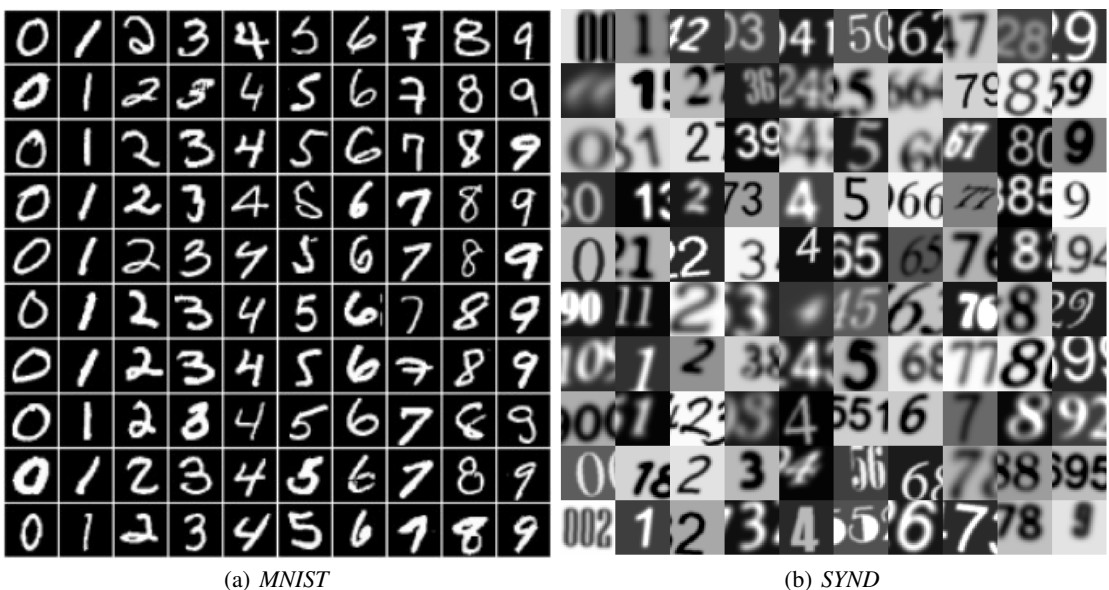

(a) *MNIST*                    (b) *SYND*

Figure 6: Visualization of *MNIST* and *SYND*.

# B    TRANSITION MATRIX $Q$

Precise definitions of Symmetry flipping and Pair flipping are presented below, where $\rho$ is the noise rate and $K$ is the number of labels.

$$\text{Symmetry flipping:} \quad Q = \begin{bmatrix} 1-\rho & \frac{\rho}{K-1} & \cdots & \frac{\rho}{K-1} & \frac{\rho}{K-1} \\ \frac{\rho}{K-1} & 1-\rho & \frac{\rho}{K-1} & \cdots & \frac{\rho}{K-1} \\ \vdots & & \ddots & & \vdots \\ \frac{\rho}{K-1} & \cdots & \frac{\rho}{K-1} & 1-\rho & \frac{\rho}{K-1} \\ \frac{\rho}{K-1} & \frac{\rho}{K-1} & \cdots & \frac{\rho}{K-1} & 1-\rho \end{bmatrix},$$

$$\text{Pair flipping:} \quad Q = \begin{bmatrix} 1-\rho & \rho & 0 & \cdots & 0 \\ 0 & 1-\rho & \rho & & 0 \\ \vdots & & \ddots & \ddots & \vdots \\ 0 & & & 1-\rho & \rho \\ \rho & 0 & \cdots & 0 & 1-\rho \end{bmatrix}.$$

Following Han et al. (2018); Jiang et al. (2018), we can corrupt clean-label datasets manually using the noise transition matrix $Q$ .

# C    DATASETS VISUALIZATION

Figure 6 shows datasets: *MNIST* and *SYND*. Figure 7 shows datasets: *Bing*, *Caltech256*, *Imagenet* and *SUN* (taking "horse" as the common class).

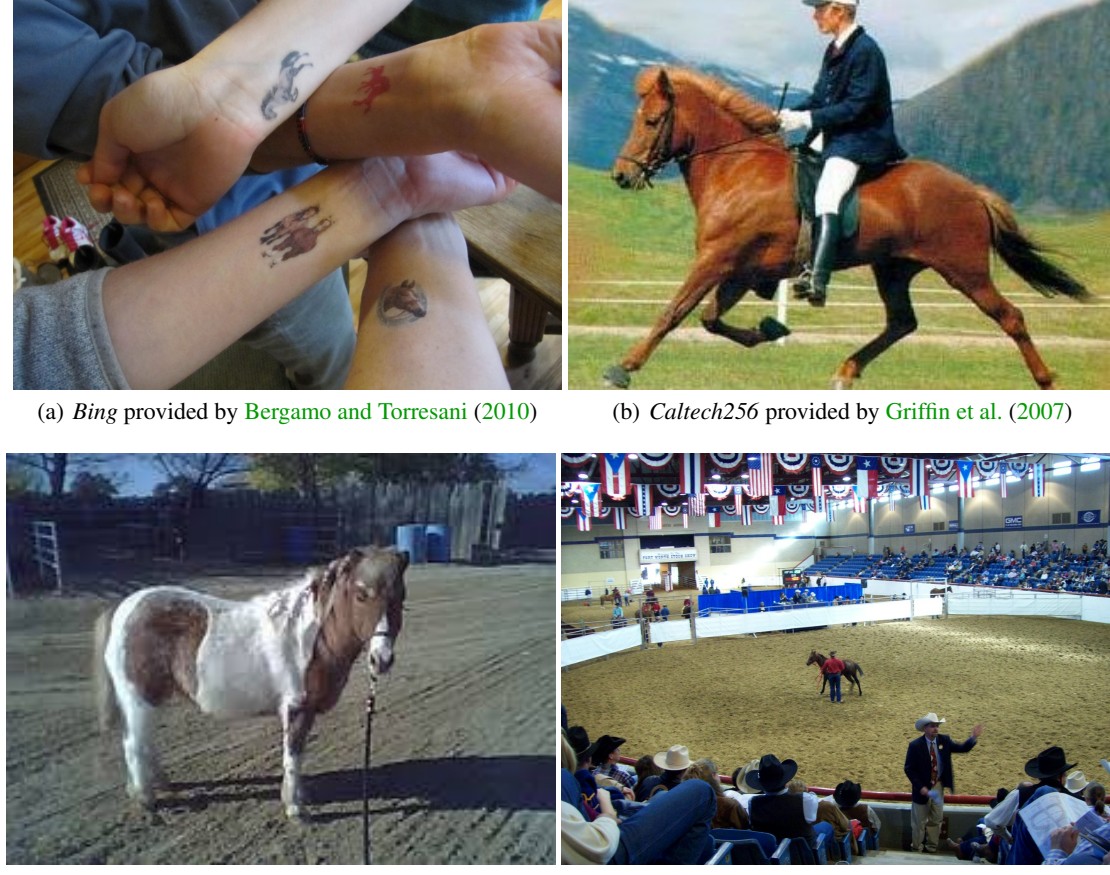

(a) *Bing* provided by Bergamo and Torresani (2010)  (b) *Caltech256* provided by Griffin et al. (2007)

(c) *ImageNet* provided by Deng et al. (2009)  (d) *SUN* provided by Xiao et al. (2010)

Figure 7: Visualization of *Bing*, *Caltech256*, *ImageNet* and *SUN* (taking "horse" as the common class).

## D  THEORETICAL ANALYSIS

This section presents some interesting theoretical findings related to WUDA problem. We use following notations in this section: 1) a space $\mathcal{X} \subset \mathbb{R}^d$ and $\mathcal{Y} = \{1, 2, \ldots, K\}$ as a label set; 2) $\tilde{p}_s(x_s, \tilde{y}_s)$, $p_s(x_s, y_s)$ and $q_s(x_s, y_s)$ represent densities of noisy, correct and incorrect multivariate random variables (m.r.v.) defined on $\mathcal{X} \times \mathcal{Y}$, respectively[1], and $\tilde{p}_{x_s}(x_s)$, $p_{x_s}(x_s)$ and $q_{x_s}(x_s)$ are their marginal densities; and 3) $p_{x_t}(x_t)$ represents density of m.r.v. $x_t$ defined on $\mathcal{X}$; and 4) we use $\ell(h(x), h'(x))$ to represent loss function between two labelling functions; and 5) we use $\tilde{R}_s(h) = \mathbb{E}_{\tilde{p}_s(x_s, \tilde{y}_s)}[\ell(h(x_s), \tilde{y}_s)]$ and $R_s(h) = \mathbb{E}_{p_s(x_s, y_s)}[\ell(h(x_s), y_s)]$ to represent expected risks on the noisy and correct m.r.v.; and 6) we use $\tilde{R}_s(h, h') = \mathbb{E}_{\tilde{p}_{x_s}(x_s)}[\ell(h(x_s), h'(x_s))]$, $R_s(h, h') = \mathbb{E}_{p_{x_s}(x_s)}[\ell(h(x_s), h'(x_s))]$ and $R_t(h, h') = \mathbb{E}_{p_{x_t}(x_t)}[\ell(h(x_t), h'(x_t))]$ to represent expected

---

[1]There are two common ways to express the density of noisy m.r.v. (see Appendix A). One way is to use a mixture of densities of correct and incorrect m.r.v..

discrepancy between two labelling functions $h, h'$ under different marginal densities; 7) the ground-truth and pseudo labeling function of the target domain are denoted by $f_t(x_t)$ and $\tilde{f}_t(x_t)$.

### D.1 WUDA RUINS UDA METHODS

Theoretically, we analyze why existing UDA methods cannot well transfer useful knowledge from noisy source data $\tilde{D}_s$ to unlabelled target data $D_t$ directly. We first present a theorem to show relations between $R_s(h)$ and $\tilde{R}_s(h)$.

**Theorem 1.** *For any labelling function $h : \mathcal{X} \to \mathcal{Y}$, if $\tilde{p}_s(x_s, \tilde{y}_s)$ is generated by a transition matrix $Q$ as demonstrated in Appendix A.1, we have*

$$\tilde{R}_s(h) = R_s(h) + \mathbb{E}_{p_{x_s}(x_s)}[\boldsymbol{\eta}^T(x_s)(Q - I)\boldsymbol{\ell}(h(x_s))], \tag{4}$$

*where $\boldsymbol{\ell}(h(x_s)) = [\ell(h(x_s), 1), ..., \ell(h(x_s), K)]^T$ and $\boldsymbol{\eta}(x_s) = [p_{Y_s|X_s}(1|x_s), ..., p_{Y_s|X_s}(K|x_s)]^T$. If $\tilde{p}_s(x_s, \tilde{y}_s)$ is generated by sample selection as described in in Appendix A.2, we have*

$$\tilde{R}_s(h) = (1 - \rho)R_s(h) + \rho\mathbb{E}_{q_{x_s}(x_s)}[\boldsymbol{\eta}_{\boldsymbol{q}}^T(x_s)\boldsymbol{\ell}(h(x_s))], \tag{5}$$

*where $\boldsymbol{\eta}_{\boldsymbol{q}}(x_s) = [q_{Y_s|X_s}(1|x_s), ..., q_{Y_s|X_s}(K|x_s)]^T$.*

**Remark 2.** In Eq. (5), $\mathbb{E}_{q_{x_s}(x_s)}[\boldsymbol{\eta}_{\boldsymbol{q}}^T(x_s)\boldsymbol{\ell}(h(x_s))]$ represents the expected risk on the incorrect m.r.v.. To ensure to obtain useful knowledge from $\tilde{P}_s$, we need to avoid $\tilde{R}_s(h) \approx \mathbb{E}_{q_{x_s}(x_s)}[\boldsymbol{\eta}_{\boldsymbol{q}}^T(x_s)\boldsymbol{\ell}(h(x_s))]$. Specifically, we assume: there is a constant $0 < M_s < \infty$ such that $\mathbb{E}_{q_{x_s}(x_s)}[\boldsymbol{\eta}_{\boldsymbol{q}}^T(x_s)\boldsymbol{\ell}(h(x_s))] \le R_s(h) + M_s$.

Theorem 1 shows that the expected risk $\tilde{R}_s(h)$ only equals $R_s(h)$ when two cases happen: 1) $Q = I$ and $\rho = 0$ and 2) some special combinations (e.g., special $p_{x_s}$, $q_{x_s}$, $Q$, $\eta$ and $\ell$) to make the second term in Eq. (4) equal zero or to make the second term in Eq. (5) equal $\rho R_s(h)$. Case 1) means that data in source domain is clean, which is not real in the wild. Case 2) almost never happens, since it is hard to find such special combinations when $p_{x_s}$, $q_{x_s}$, $Q$ and $\eta$ are unknown. Thus, $\tilde{R}_s(h)$ has an essential difference with $R_s(h)$. Then, following proof skills in Ben-David et al. (2010), we derive the upper bound of $R_t(h)$ as follows.

**Theorem 2.** *For any labelling function $h : \mathcal{X} \to \mathcal{Y}$, we have*

$$R_t(h, f_t) \le \underbrace{\tilde{R}_s(h)}_{(i) \ \textit{risk on noisy data}} + \underbrace{|R_t(h, \tilde{f}_t) - \tilde{R}_s(h, \tilde{f}_t)|}_{(ii) \ \textit{discrepancy between distributions}} + \underbrace{|R_s(h, \tilde{f}_t) - R_s(h)|}_{(iii) \ \textit{domain dissimilarity}}$$

$$+ \underbrace{|\tilde{R}_s(h) - R_s(h)| + |\tilde{R}_s(h, \tilde{f}_t) - R_s(h, \tilde{f}_t)|}_{(iv) \ \textit{noise effects from source } \Delta_s} + \underbrace{|R_t(h, f_t) - R_t(h, \tilde{f}_t)|}_{(v) \ \textit{noise effects from target } \Delta_t}. \tag{6}$$

**Remark 3.** To ensure that we can gain useful knowledge from $\tilde{f}_t(x_t)$, we assume: there is a constant $0 < M_t < \infty$ such that $\mathbb{E}_{q_{x_s}(x)}[\ell(h(x), \tilde{f}_t(x))] \le R_s(h, \tilde{f}_t) + M_t$ and $\mathbb{E}_{q_{x_t}(x)}[\ell(h(x), \tilde{f}_t(x))] \le R_t(h, f_t) + M_t$, where $q_{x_t}(x) = p_{x_t}(x)1_A(x)/P_{x_t}(A)$ and $A = \{x : \tilde{f}_t(x) \ne f_t(x)\}$.

It is clear that the upper bound of $R_t(h, f_t)$, shown in Eq. (6), has 5 components. However, existing UDA methods only focus on minimizing $(i) + (ii)$ (Ganin et al., 2016; Ghifary et al., 2017; Long et al., 2015) or $(i) + (ii) + (iii)$ (Saito et al., 2017), which ignores terms $(iv)$ and $(v)$ (i.e., $\Delta = \Delta_s + \Delta_t$). Thus, in theory, existing UDA methods cannot handle wildly unsupervised domain adaptation well.

### D.2 TWO-STEP APPROACH IS A COMPROMISE SOLUTION

To reduce noise effects, a straightforward solution is two-step approach. In the first step, we can train a classifier with noisy source data using Co-teaching (Han et al., 2018) and use this classifier to annotate pseudo

labels for source data. In the second step, we use ATDA (Saito et al., 2017) to train a target-domain classifier with pseudo-label-source and target data.

Nonetheless, the pseudo-labeled source data is still noisy. Let labels of noisy source data $\tilde{y}_s$ be replaced with pseudo labels $\tilde{y}_s'$ after pre-processing. Noise effects $\Delta$ will become pseudo-label effects $\Delta_p$ as follows.

$$\Delta_p = \underbrace{|\tilde{R}_s'(h) - R_s(h)| + |\tilde{R}_s'(h, \tilde{f}_t) - R_s(h, \tilde{f}_t)|}_{\text{pseudo-label-source effects } \Delta_s'} + \Delta_t, \tag{7}$$

where $\tilde{R}_s'(h)$ and $\tilde{R}_s'(h, \tilde{f}_t)$ correspond to $\tilde{R}_s(h)$ and $\tilde{R}_s(h, \tilde{f}_t)$ in $\Delta_s$. It is clear that the difference between $\Delta_p$ and $\Delta$ is $\Delta_s' - \Delta_s$. The first term in $\Delta_s'$ may be less than that in $\Delta_s$ due to Co-teaching, but the second term in $\Delta_s'$ may be higher than that in $\Delta_s$ since Co-teaching does not consider to minimize it. Thus, it is hard to say whether $\Delta_s' < \Delta_s$ (i.e., $\Delta_p < \Delta$). This means that two-step approach may not really reduce noise effects.

### D.3 WHY DOES BUTTERFLY CAN ELIMINATE NOISE EFFECT?

To eliminate noise effects $\Delta$, we aim to select correct data simultaneously from noisy source data and pseudo-labeled target data. In theory, we prove that noise effects will be eliminated if we can select correct data with a high probability. Let $\rho_{01}^s$ represent the probability that incorrect data is selected from noisy source data, and $\rho_{01}^t$ represent the probability that incorrect data is selected from pseudo-labeled target data. Theorem 3 shows that $\Delta \to 0$ if $\rho_{01}^s \to 0$ and $\rho_{01}^t \to 0$ and presents a new upper bound of $R_t(h, f_t)$.

**Theorem 3.** *Given two m.r.v. $(X_s, Y_s, U_s)$ defined on $\mathcal{X} \times \mathcal{Y} \times \mathcal{V}$ and $(X_t, U_t)$ defined on $\mathcal{X} \times \mathcal{V}$, under the assumptions in Remarks 2 and 3, $\forall \epsilon \in (0, 1)$, there are $\delta_s$ and $\delta_t$, if $\mathbb{E}_{p_{x_t}'(x_t)}[\ell(h(x_t), f_t(x_t))] \leq R_t(h, f_t) + \rho_{01}^s M_t$, $\rho_{01}^s < \delta_s$ and $\rho_{01}^t < \delta_t$, for any labeling function $h$, we will have*

$$|\tilde{R}_s^{po}(h, \tilde{f}_t, u_s) - R_s(h, \tilde{f}_t)| + |\tilde{R}_s^{po}(h, u_s) - R_s(h)| < 2\epsilon. \tag{8}$$

*Moreover, if $\rho_{01}^s \leq \delta_s$ and $\rho_{01}^t \leq \delta_t$, we will have*

$$R_t(h, f_t) \leq \underbrace{\tilde{R}_s^{po}(h, u_s)}_{(i) \text{ risk on noisy data}} + \underbrace{|\tilde{R}_t^{po}(h, \tilde{f}_t, u_t) - \tilde{R}_s^{po}(h, \tilde{f}_t, u_s)|}_{(ii) \text{ discrepancy between distributions}} + \underbrace{|R_s(h, \tilde{f}_t) - R_s(h)|}_{(iii) \text{ domain dissimilarity}}$$

$$+ \underbrace{2\epsilon}_{(iv) \text{ noise effects from source } \Delta_s} + \underbrace{2\epsilon}_{(v) \text{ noise effects from target } \Delta_t}, \tag{9}$$

*where $p_{x_t}'(x) = p_{x_t}(x)1_B(x)/P_{x_t}(B)$, $\tilde{R}_s^{po}(h, u_s) = (1 - \rho_{u_s})^{-1}\mathbb{E}_{\tilde{p}_s^{po}(x_s, y_s, u_s)}[u_s\ell(h(x_s), y_s)]$, $\tilde{R}_t^{po}(h, \tilde{f}_t, u_t) = (1 - \rho_{u_t})^{-1}\mathbb{E}_{\tilde{p}_t^{po}(x_t, u_t)}[u_t\ell(h(x_t), \tilde{f}_t(x_t))]$, $\tilde{R}_s^{po}(h, \tilde{f}_t, u_s) = (1 - \rho_{u_s})^{-1}\mathbb{E}_{\tilde{p}_s^{po}(x_s, y_s, u_s)}[u_s\ell(h(x_s), \tilde{f}_t(x_s))]$, $\tilde{p}_s^{po}(x_s, y_s, u_s)$ is the density of $(X_s, Y_s, U_s)$, $\tilde{p}_t^{po}(x_t, u_t)$ is the density of $(X_t, U_t)$, $\rho_{u_s} = \int_{\mathcal{X}} \sum_{y_s=1}^K \tilde{p}_s^{po}(x_s, y_s, 0)dx_s < 1$, $\rho_{u_t} = \int_{\mathcal{X}} \tilde{p}_t^{po}(x_t, 0)dx_t < 1$, $B = \mathcal{X}/A$ and $\mathcal{V} = \{0, 1\}$.*

**Remark 4.** In Appendix H.3.1, we give precise definitions of $\rho_{01}^s$ and $\rho_{01}^t$ and demonstrate the meaning of $\mathbb{E}_{p_{x_t}'(x_t)}[\ell(h(x_t), f_t(x_t))] \leq R_t(h, f_t) + \rho_{01}^s M_t$ (Remark 5).

Data drawn from the distribution of $(X_s, Y_s, U_s)$ can be regarded as a pool that mixes the selected ($u_s = 1$) and unselected ($u_s = 0$) noisy source data. Data drawn from the distribution of $(X_t, U_t)$ can be regarded as a pool that mixes the selected ($u_t = 1$) and unselected ($u_t = 0$) pseudo-labeled target data. Theorem 3 shows that if selected data have a high probability to be correct ones ($\rho_{01}^s \to 0$ and $\rho_{01}^t \to 0$), then $\Delta_s$ and $\Delta_t$ approach 0, meaning that noise effects are eliminated. This motivates us to find a reliable way to select correct data from noisy source data and pseudo-labeled target data and build up a one-step approach for WUDA.

**Why Butterfly?**    Guided by Theorem 3, a robust approach should check high-correctness data out (meaning $\rho_{01}^s \to 0$ and $\rho_{01}^t \to 0$). This checking process will make $(iv)$ and $(v)$, $2\epsilon + 2\epsilon$, become $0$. Then, we can obtain gradients of $\tilde{R}_s^{\mathrm{po}}(h, u_s)$, $\tilde{R}_s(h, \tilde{f}_t, u_s)$ and $\tilde{R}_t^{\mathrm{po}}(h, \tilde{f}_t, u_t)$ w.r.t. parameters of $h$ and use these gradients to minimize them, which minimizes $(i)$ and $(ii)$ as $(i) + (ii) \leq \tilde{R}_s^{\mathrm{po}}(h, u_s) + \tilde{R}_s(h, \tilde{f}_t, u_s) + \tilde{R}_t^{\mathrm{po}}(h, \tilde{f}_t, u_t)$. Note that $(iii)$ cannot be directly minimized since we cannot pinpoint clean source data. However, following Saito et al. (2017), we can indirectly minimize $(iii)$ via minimizing $\tilde{R}_s^{\mathrm{po}}(h, u_s) + \tilde{R}_s^{\mathrm{po}}(h, \tilde{f}_t, u_s)$, as $(iii) \leq R_s(h, \tilde{f}_t) + R_s(h) \leq \tilde{R}_s^{\mathrm{po}}(h, u_s) + \tilde{R}_s^{\mathrm{po}}(h, \tilde{f}_t, u_s) + 2\epsilon$, where the last inequality follows Eq. (8). This means that a robust approach guided by Theorem 3 can minimize all terms in the right side of inequality in Eq. (9).

To realize this robust approach, we propose a Butterfly framework (Algorithm 1), which trains four networks dividing into two branches (Figure 3). By using dual-checking principle, Branch-I checks which data is correct in the mixture domain; while Branch-II checks which pseudo-labeled target data is correct. To ensure these checked data highly-correct, we apply the small-loss trick based on memorization effects of deep learning Arpit et al. (2017). After cross-propagating these checked data Bengio (2014), Butterfly can obtain high-quality DIR and TSR simultaneously in an iterative manner. Theoretically, Branch-I minimizes $(i) + (ii) + (iii) + (iv)$; while Branch-II minimizes $(ii) + (v)$. This means that Butterfly can minimize all terms in the right side of inequality in Eq. (9). Note that empirical estimators of $\tilde{R}_s^{\mathrm{po}}(h, u_s)$, $\tilde{R}_t^{\mathrm{po}}(h, \tilde{f}_t, u_t)$ and $\tilde{R}_s^{\mathrm{po}}(h, \tilde{f}_t, u_s)$ (in Theorem 3) can be expressed using the loss function of Butterfly (see Eq. (1)).

**Relations to Co-teaching.**    As Butterfly is related to Co-teaching, we discuss their major differences here. Although Co-teaching (Han et al., 2018) applies the small-loss trick and the cross-update technique to train deep networks against noisy data, it can only deal with one-domain problem instead cross-domain problem. Besides, we argue that Butterfly is not a simple mixtrue of Co-teaching and ATDA for two reasons.

First, network structure of Butterfly is different with that of ATDA and Co-teaching: Butterfly maintains four networks; while ATDA maintains three and Co-teaching maintains two. We cannot simply combine ADTA and Co-teaching to derive Butterfly. Second, we have justified that the sequential mixture of Co-teaching and ATDA (i.e., two-step method) cannot eliminate noise effects caused by noisy source data (see Section 4). Specifically, two-step methods only take care of part of noise effects but Butterfly takes care of the whole noise effects. Thus, Butterfly is the first method to eliminate noise effects rather than alleviate it.

**Relations to TCL.**    Recently, *transferable curriculum learning* (TCL) is a robust UDA method to handle noise Shu et al. (2019). TCL uses small-loss trick to train the *domain-adversarial neural network* (DANN) Ganin et al. (2016). However, TCL can only minimize $(i) + (ii) + (iv)$, while Butterfly can minimize all terms in the right side of Eq. (9).

# E    RESULTS ON HUMAN-SENTIMENT WUDA TASKS

Tables 5 and 6 report the target-domain accuracy of each method for 24 human-sentiment WUDA tasks. For these tasks, B-Net has the highest average accuracy. It should be noted that two-step method does not always perform better than existing UDA methods, such as for $20\%$-noise situation. The main reason is Co-teaching performs poorly when pinpointing clean source data from noisy source data. Another observation is that noise effects is not eliminated like classification results on digit WUDA tasks. The main reason is that these datasets provide fixed features and we cannot extract better features in the training process. However, in digit WUDA tasks, we can gradually obtain better features for each domain and finally eliminate noise effects.

Table 5: Target-domain accuracy on 12 human-sentiment WUDA tasks with the 20% noise rate. Bold values mean the highest values in each row.

| Tasks | DAN | DANN | ATDA | TCL | Co+TCL | Co+ATDA | B-Net |
|---|---|---|---|---|---|---|---|
| $B{\to}D$ | 68.28% | 68.08% | 70.31% | 71.40% | 67.81% | 66.70% | **71.84%** |
| $B{\to}E$ | 63.78% | 63.53% | 72.79% | 65.08% | 60.54% | 68.89% | **75.92%** |
| $B{\to}K$ | 65.48% | 64.63% | 71.79% | 66.80% | 61.23% | 66.51% | **76.32%** |
| $D{\to}B$ | 64.63% | 64.52% | 70.25% | 67.33% | 65.22% | 68.04% | **70.56%** |
| $D{\to}E$ | 65.33% | 65.16% | 69.99% | 66.74% | 64.55% | 67.32% | **73.73%** |
| $D{\to}K$ | 65.68% | 66.28% | 74.53% | 68.82% | 67.98% | 72.20% | **77.97%** |
| $E{\to}B$ | 60.41% | 60.15% | **63.89%** | 63.13% | 61.18% | 61.08% | 62.22% |
| $E{\to}D$ | 62.35% | 61.67% | 62.30% | 62.93% | 60.81% | 59.77% | **63.53%** |
| $E{\to}K$ | 72.05% | 71.51% | 74.00% | 75.36% | 72.65% | 70.85% | **78.96%** |
| $K{\to}B$ | 59.94% | 59.40% | **63.53%** | 62.77% | 60.71% | 61.22% | 63.36% |
| $K{\to}D$ | 61.46% | 61.51% | 64.66% | 64.16% | 64.15% | 64.94% | **66.98%** |
| $K{\to}E$ | 70.60% | 72.23% | 74.75% | 74.14% | 68.95% | 69.69% | **76.96%** |
| Average | 65.00% | 64.89% | 69.40% | 67.39% | 64.65% | 66.43% | **71.53%** |

Table 6: Target-domain accuracy on 12 human-sentiment WUDA tasks with the 45% noise rate. Bold values mean the highest values in each row.

| Tasks | DAN | DANN | ATDA | TCL | Co+TCL | Co+ATDA | B-Net |
|---|---|---|---|---|---|---|---|
| $B{\to}D$ | 52.43% | 52.98% | 53.56% | 54.44% | 53.21% | 54.32% | **56.59%** |
| $B{\to}E$ | 52.17% | 53.50% | 55.14% | 54.14% | 53.98% | **57.34%** | 55.74% |
| $B{\to}K$ | 52.89% | 51.84% | 51.14% | 53.32% | 51.77% | 53.28% | **57.00%** |
| $D{\to}B$ | 53.11% | 53.04% | 54.48% | 53.27% | 54.85% | **55.95%** | 55.15% |
| $D{\to}E$ | 51.30% | 53.04% | 54.21% | 53.77% | 55.63% | 56.08% | **58.91%** |
| $D{\to}K$ | 52.15% | 53.17% | 57.99% | 52.45% | 58.10% | 59.94% | **66.20%** |
| $E{\to}B$ | 51.38% | 51.08% | 52.54% | 52.14% | 54.88% | 53.30% | **54.93%** |
| $E{\to}D$ | 52.83% | 51.24% | 49.02% | 52.57% | 50.03% | 49.62% | **52.88%** |
| $E{\to}K$ | 54.21% | 53.58% | 51.66% | 55.04% | 56.15% | 52.10% | **56.12%** |
| $K{\to}B$ | 50.44% | 51.77% | **51.96%** | 51.50% | 53.81% | 52.59% | 51.39% |
| $K{\to}D$ | 52.20% | 51.45% | 52.86% | 53.19% | 55.69% | 54.52% | **53.53%** |
| $K{\to}E$ | **54.72%** | 53.33% | 52.11% | 53.46% | 51.26% | 52.62% | 53.71% |
| Average | 52.49% | 52.50% | 53.65% | 53.27% | 54.11% | 54.31% | **56.01%** |

# F EXPERIMENTAL SETTINGS

## F.1 NETWORK STRUCTURE AND OPTIMIZER

We implement all methods on Python 3.6 with a NIVIDIA P100 GPU. We use MomentumSGD for optimization in digit and real-world tasks, and set the momentum as 0.9. We use Adagrad for optimization in human-sentiment tasks because of sparsity of review data (Saito et al., 2017). $F_1$, $F_2$, $F_{t1}$ and $F_{t2}$ are 6-layer CNN (3 convolutional layers and 3 fully-connected layers) for digit tasks; and are 3-layer neural networks (3 fully-connected layers) for human-sentiment tasks; and are 4-layer neural networks (4 fully-connected layers) for real-world tasks. The ReLU active function is used as avtivation function of these networks. Besides, dropout and batch normalization are also used. The network topology is shown in Figures 8, 9 and 10. As deep networks are highly nonconvex, even with the same network and optimization method, different initializations can lead to different local optimal. Thus, following Malach and Shalev-Shwartz (2017), we also take four networks with the same architecture but different initializations as four classifiers.

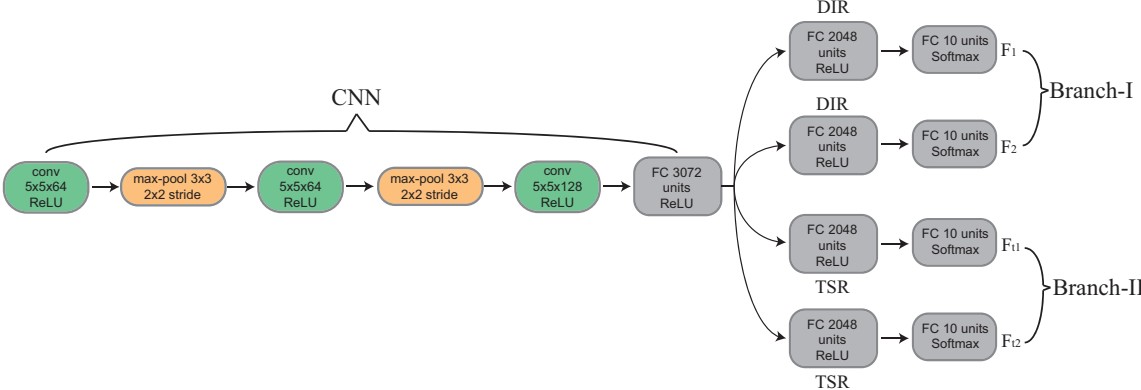

Figure 8: The architecture of B-Net for digit WUDA tasks *SYND ↔ MNIST*. We added BN layer in the last convolution layer in CNN and FC layers in $F_1$ and $F_2$. We also used dropout in the last convolution layer in CNN and FC layers in $F_1$, $F_2$, $F_{t1}$ and $F_{t2}$ (dropout probability is set to $0.5$).

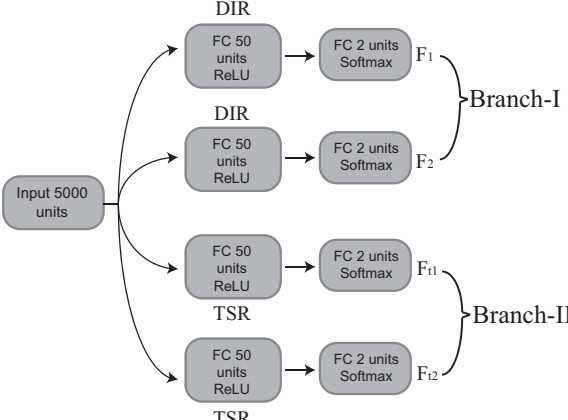

Figure 9: The architecture of B-Net for human-sentiment WUDA tasks. We added BN layer in the first FC layers in $F_1$ and $F_2$. We also used dropout in the first FC layers in $F_1$, $F_2$, $F_{t1}$ and $F_{t2}$ (dropout probability is set to $0.5$).

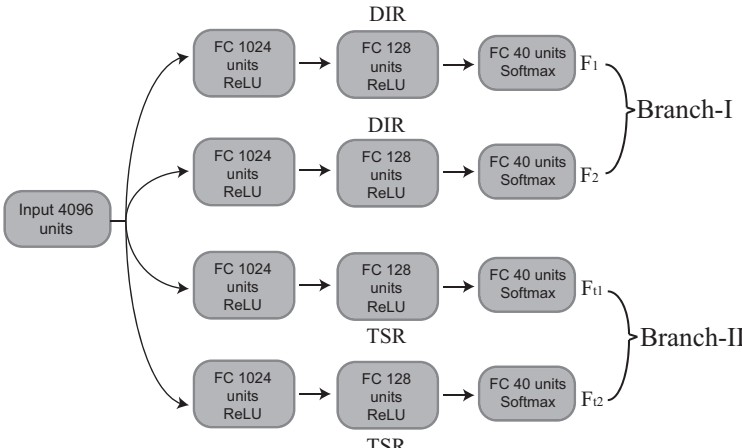

Figure 10: The architecture of B-Net for real-world WUDA tasks. We added BN layer in the first FC layers in $F_1$, $F_2$, $F_{t1}$ and $F_{t2}$. We also used dropout in the first FC layers in $F_1$, $F_2$, $F_{t1}$ and $F_{t2}$ (dropout probability is set to $0.5$).

## F.2 EXPERIMENTAL SETUP

For all 35 WUDA tasks and 2 UDA tasks, $T_k$ is set to 5, $T_{max}$ is set to 30. Learning rate is set to $0.01$ for simulated tasks and $0.05$ for real-world WUDA tasks and UDA tasks, $\gamma_t$ is set to $0.05$ for simulated tasks and $0.02$ for real-world WUDA tasks and UDA tasks. Confidence level of labelling function in line 8 of Algorithm 1 is set to $0.95$ for 8 digit tasks and 2 UDA tasks, and $0.9$ for 24 human-sentiment tasks and $0.8$ for real-world WUDA tasks. $\gamma$ is set to $0.4$ for digit tasks, $0.1$ for human-sentiment tasks, $0.2$ for real-world WUDA tasks and $0.1$ for UDA tasks. $n_{t,max}^l$ is set to $15,000$ for digit tasks and UDA tasks, $500$ for human-sentiment tasks and $4000$ for real-world WUDA tasks. $N_{max}$ is set to $1000$ for digit tasks and UDA tasks, and $200$ for human-sentiment and real-world tasks. Batch size is set to $128$ for digit, real-world WUDA tasks and UDA tasks, and $24$ for human-sentiment tasks. Penalty parameter is set to $0.01$ for digit, real-world WUDA tasks and UDA tasks, and $0.001$ for human-sentiment tasks.

To fairly compare all methods, they have the same network structure. Namely, ATDA, DAN, DANN, TCL and B-Net adopt the same network structure for each dataset. Note that DANN and TCL use the same structure for their discriminate networks. All experiments are repeated ten times and we report the average accuracy value and *standard deviation* (STD) of accuracy values of ten experiments.

## F.3 LINKS TO DATASETS

*Digit* datasets (*MNIST* and *SYN Digit (SYND)*) can be downloaded from official code of ATDA. The link is https://github.com/ksaito-ut/atda.

*Sentiment* datasets (*Amazon products reviews*) can be downloaded from the official code of marginalized Stacked Denoising Autoencoder (mSDA). The link is https://www.cse.wustl.edu/~mchen/code/mSDA.tar.

*Real-world* datasets (*BCIS*) can be downloaded from the website of the project "A Testbed for Cross-Dataset Analysis": https://sites.google.com/site/crossdataset/home/files ("setup DENSE decaf7", 1.3GB, decaf7 features).

## G  RUNNING TIME

Table 7 shows the average running time of each method on the task *SYND→MNIST*. Although B-Net trains four networks, its running time is still comparable to most baselines.

Table 7: Running time for each method on the task *SYND→MNIST* (minutes).

| Methods | DAN | DANN | ATDA | TCL | Co+ATDA | B-Net |
|---------|-----|------|------|-----|---------|-------|
| Time | 17.17 | 9.02 | 17.17 | 14.04 | 18.28 | 20.55 |

## H  PROOFS

This section provides proofs of theorems demonstrated in the supplementary.

### H.1  PROOF OF THEOREM 1

*Proof.* We will fist prove Eq. (4) (Case 1) and then prove Eq. (5) (Case 2).

**Case** 1. According to definition of $\tilde{R}_s(h)$, we have

$$
\begin{aligned}
\tilde{R}_s(h) &= \mathbb{E}_{\tilde{p}_s(x_s,\tilde{y}_s)}[\ell(h(x_s),\tilde{y}_s)] \\
&= \int_{\mathcal{X}} \sum_{\tilde{y}_s=1}^{K} \ell(h(x_s),\tilde{y}_s)\tilde{p}_s(x_s,\tilde{y}_s)dx_s \\
&= \int_{\mathcal{X}} \sum_{\tilde{y}_s=1}^{K} \ell(h(x_s),\tilde{y}_s)\tilde{p}_{\tilde{Y}_s|X_s}(\tilde{y}_s|x_s)p_{x_s}(x_s)dx_s \\
&= \int_{\mathcal{X}} \tilde{\boldsymbol{\eta}}^T(x_s)\boldsymbol{\ell}(h(x_s))p_{x_s}(x_s)dx_s,
\end{aligned}
\tag{10}
$$

where $\boldsymbol{\ell}(h(x_s)) = [\ell(h(x_s),1),...,\ell(h(x_s),K)]^T$ and $\tilde{\boldsymbol{\eta}}(x_s) = [\tilde{p}_{\tilde{Y}_s|X_s}(1|x_s),\ldots,\tilde{p}_{\tilde{Y}_s|X_s}(K|x_s)]^T$. According to definition of the transition matrix $Q$, we know that

$$
\tilde{\boldsymbol{\eta}}^T(x_s) = \boldsymbol{\eta}^T(x_s)Q,
\tag{11}
$$

where $\boldsymbol{\eta}(x_s) = [p_{Y_s|X_s}(1|x_s),\ldots,p_{Y_s|X_s}(K|x_s)]^T$. Substituting Eq. (11) into Eq. (10), we have

$$
\begin{aligned}
\tilde{R}_s(h) &= \int_{\mathcal{X}} \boldsymbol{\eta}^T(x_s)Q\boldsymbol{\ell}(h(x_s))p_{x_s}(x_s)dx_s \\
&= \int_{\mathcal{X}} \boldsymbol{\eta}^T(x_s)I\boldsymbol{\ell}(h(x_s))p_{x_s}(x_s)dx_s + \int_{\mathcal{X}} \boldsymbol{\eta}^T(x_s)(Q-I)\boldsymbol{\ell}(h(x_s))p_{x_s}(x_s)dx_s \\
&= R_s(h) + \mathbb{E}_{p_{x_s}(x_s)}[\boldsymbol{\eta}^T(x_s)(Q-I)\boldsymbol{\ell}(h(x_s))].
\end{aligned}
$$

Hence, Case 1 is proved.

**Case** 2. According to definition of $\tilde{R}_s(h)$ and Eq. (3), we have

$$
\begin{aligned}
\tilde{R}_s(h) &= \mathbb{E}_{\tilde{p}_s(x_s,\tilde{y}_s)}[\ell(h(x_s),\tilde{y}_s)] \\
&= \int_{\mathcal{X}} \sum_{\tilde{y}_s=1}^{K} \ell(h(x_s),\tilde{y}_s)\tilde{p}_s(x_s,\tilde{y}_s)dx_s \\
&= \int_{\mathcal{X}} \sum_{y_s=1}^{K} \ell(h(x_s),y_s)\big((1-\rho)p_s(x_s,y_s)+\rho q_s(x_s,y_s)\big)dx_s \\
&= (1-\rho)\int_{\mathcal{X}} \sum_{y_s=1}^{K} \ell(h(x_s),y_s)p_s(x_s,y_s)dx_s + \rho\int_{\mathcal{X}} \sum_{y_s=1}^{K} \ell(h(x_s),y_s)q_s(x_s,y_s)dx_s \\
&= (1-\rho)R_s(h) + \rho\int_{\mathcal{X}} \sum_{y_s=1}^{K} \ell(h(x_s),y_s)q_{Y_s|X_s}(y_s|x_s)q_{x_s}(x_s)dx_s.
\end{aligned}
\tag{12}
$$

Let $\boldsymbol{\eta_q}(x_s) = [q_{Y_s|X_s}(1|x_s),...,q_{Y_s|X_s}(K|x_s)]^T$, we have

$$
\tilde{R}_s(h) = (1-\rho)R_s(h) + \rho\mathbb{E}_{q_{x_s}(x_s)}[\boldsymbol{\eta_q}^T(x_s)\boldsymbol{\ell}(h(x_s))].
$$

Hence, Case 2 is proved. $\qquad\square$

### H.2 PROOF OF THEOREM 2

*Proof.* For any labelling function $h$, we have

$$
\begin{aligned}
R_t(h,f_t) &= R_t(h,f_t) + \tilde{R}_s(h) - \tilde{R}_s(h) + R_s(h,f_t) - R_s(h,f_t) \\
&= \tilde{R}_s(h) + R_t(h,f_t) - \tilde{R}_s(h,f_t) + R_s(h,f_t) - R_s(h) + R_s(h) - \tilde{R}_s(h) \\
&\quad + \tilde{R}_s(h,f_t) - R_s(h,f_t).
\end{aligned}
\tag{13}
$$

Since we do not know $f_t$, we substitute following equations into Eq. (13),

$$
\begin{aligned}
R_t(h,f_t) &= R_t(h,\tilde{f}_t) + R_t(h,f_t) - R_t(h,\tilde{f}_t), \\
\tilde{R}_s(h,f_t) &= \tilde{R}_s(h,\tilde{f}_t) + \tilde{R}_s(h,f_t) - \tilde{R}_s(h,\tilde{f}_t), \\
R_s(h,f_t) &= R_s(h,\tilde{f}_t) + R_s(h,f_t) - R_s(h,\tilde{f}_t).
\end{aligned}
$$

Then, we have

$$
\begin{aligned}
R_t(h,f_t) &= \tilde{R}_s(h) + R_t(h,\tilde{f}_t) - \tilde{R}_s(h,\tilde{f}_t) + R_s(h,\tilde{f}_t) - R_s(h) \\
&\quad + R_s(h) - \tilde{R}_s(h) + \tilde{R}_s(h,\tilde{f}_t) - R_s(h,\tilde{f}_t) + R_t(h,f_t) - R_t(h,\tilde{f}_t) \\
&\leq \tilde{R}_s(h) + |R_t(h,\tilde{f}_t) - \tilde{R}_s(h,\tilde{f}_t)| + |R_s(h,\tilde{f}_t) - R_s(h)| \\
&\quad + |\tilde{R}_s(h) - R_s(h)| + |\tilde{R}_s(h,\tilde{f}_t) - R_s(h,\tilde{f}_t)| + |R_t(h,f_t) - R_t(h,\tilde{f}_t)|.
\end{aligned}
$$

Hence, this theorem is proved. $\qquad\square$

### H.3 PROOF OF THEOREM 3

#### H.3.1 PRELIMINARY

Before stating the proof, we first present a random variable below.

Let $(X_t, V_t)$ be a m.r.v. defined on $\mathcal{X} \times \mathcal{V}$ with respective a density $p_t^{\mathrm{po}}(x_t, v_t)$, where $\mathcal{V} = \{0, 1\}$. $V_t$ can be regarded as *perfect-selection random variables*. Namely, $V_t = 1$ means $f_t(x_t) = \tilde{f}_t(x_t)$ and $V_t = 0$ means $f_t(x_t) \neq \tilde{f}_t(x_t)$. Let $p_{V_t}^{\mathrm{po}}(v_t)$ be the marginal density of $p_t^{\mathrm{po}}(x_t, v_t)$. It is clear that, higher value of $p_{V_t}^{\mathrm{po}}(V_t = 1)$ means that $\tilde{f}_t$ is more like $f_t$. In following, we use $1 - \rho_{v_t}$ to represent $p_{V_t}^{\mathrm{po}}(V_t = 1)$.

Then, we will show 1) relation between $(X_s, Y_s, V_s)$ and $(X_s, Y_s, U_s)$, 2) relation between $(X_t, V_t)$ and $(X_t, U_t)$ and definitions of $\rho_{01}^s$ and $\rho_{01}^t$. Based on $(X_t, V_t)$ and $(X_s, Y_s, V_s)$ defined in Appendix A.2, the densities of $(X_s, Y_s, U_s)$ and $(X_t, U_t)$ can be expressed as follows.

$$\tilde{p}_{X_s, Y_s | U_s}^{\mathrm{po}}(x_s, y_s | i) = \rho_{0i}^s p_{X_s, Y_s | V_s}^{\mathrm{po}}(x_s, y_s | 0) + \rho_{1i}^s p_{X_s, Y_s | V_s}^{\mathrm{po}}(x_s, y_s | 1),$$

$$\tilde{p}_{X_t | U_t}^{\mathrm{po}}(x_t | i) = \rho_{0i}^t p_{X_t | V_t}^{\mathrm{po}}(x_t | 0) + \rho_{1i}^t p_{X_t | V_t}^{\mathrm{po}}(x_t | 1),$$

where $\rho_{ji}^s = \Pr(V_s = j | U_s = i)$ represents the probability of the event: $V_s = j$ given $U_s = i$, $\rho_{ji}^t = \Pr(V_t = j | U_t = i)$ represents the probability of the event: $V_t = j$ given $U_t = i$ $(i, j = 0, 1)$. Since $p_s(x_s, y_s) = p_{X_s, Y_s | V_s}^{\mathrm{po}}(x_s, y_s | 1)$, $q_s(x_s, y_s) = p_{X_s, Y_s | V_s}^{\mathrm{po}}(x_s, y_s | 0)$, $p_{X_t | V_t}^{\mathrm{po}}(x_t | 0) = p_{x_t}(x_t) 1_A(x_t) / P_{x_t}(A) = q_{x_t}(x_t)$ and $p_{X_t | V_t}^{\mathrm{po}}(x_t | 1) = p_{x_t}(x_t) 1_B(x_t) / P_{x_t}(B) = p'_{x_t}(x_t)$ $(A = \{x : \tilde{f}_t(x) \neq f_t(x)\}$, $B = \mathcal{X} / A)$, we have

$$\tilde{p}_{X_s, Y_s | U_s}^{\mathrm{po}}(x_s, y_s | i) = \rho_{0i}^s q_s(x_s, y_s) + \rho_{1i}^s p_s(x_s, y_s), \tag{14}$$

$$\tilde{p}_{X_t | U_t}^{\mathrm{po}}(x_t | i) = \rho_{0i}^t q_{x_t}(x_t) + \rho_{1i}^t p'_{x_t}(x_t). \tag{15}$$

Next, we give a lemma to show relation between $\tilde{R}_s^{\mathrm{po}}(h, u_s)$ and $R_s(h)$.

**Lemma 1.** *Given the multivariate random variable $(X_s, Y_s, U_s)$ with the probability $\tilde{p}_s^{po}(x_s, y_s, u_s)$ and Eq. (14), we have*

$$|\tilde{R}_s^{po}(h, u_s) - R_s(h)| \leq \rho_{01}^s \max\{\mathbb{E}_{q_s(x_s, y_s)}[\ell(h(x_s), y_s)], R_s(h)\}. \tag{16}$$

*Proof.* According to definition of $\tilde{R}_s^{\mathrm{po}}(h, u_s)$ in Theorem 3, we have

$$\tilde{R}_s^{\mathrm{po}}(h, u_s) = (1 - \rho_{u_s})^{-1} \int_{\mathcal{X}} \sum_{u_s=0}^1 \sum_{y_s=1}^K u_s \ell(h(x_s), y_s) \tilde{p}_s^{\mathrm{po}}(x_s, y_s, u_s) dx_s$$

$$= (1 - \rho_{u_s})^{-1} \int_{\mathcal{X}} \sum_{y_s=1}^K \ell(h(x_s), y_s) \tilde{p}_{X_s, Y_s | U_s}^{\mathrm{po}}(x_s, y_s | 1) \tilde{p}_{U_s}^{\mathrm{po}}(1) dx_s$$

$$\overset{(a)}{=} (1 - \rho_{u_s})^{-1} (1 - \rho_{u_s}) \int_{\mathcal{X}} \sum_{y_s=1}^K \ell(h(x_s), y_s) \left( \rho_{01}^s q_s(x_s, y_s) + \rho_{11}^s p_s(x_s, y_s) \right) dx_s$$

$$= \rho_{01}^s \mathbb{E}_{q_s(x_s, y_s)}[\ell(h(x_s), y_s)] + \rho_{11}^s R_s(h),$$

where $(a)$ is based on the definition of $\rho_{u_s}$ and Eq. (14). Thus, we have

$$|\tilde{R}_s^{\mathrm{po}}(h, u_s) - R_s(h)| = |\rho_{01}^s \mathbb{E}_{q_s(x_s, y_s)}[\ell(h(x_s), y_s)] - (1 - \rho_{11}^s) R_s(h)|$$

$$\leq \rho_{01}^s \max\{\mathbb{E}_{q_s(x_s, y_s)}[\ell(h(x_s), y_s)], R_s(h)\}.$$

This lemma is proved. $\qquad \square$

Similar with Lemma 1, we can obtain

$$|\tilde{R}_s^{po}(h, \tilde{f}_t, u_t) - R_s(h, \tilde{f}_t)| \leq \rho_{01}^s \max\{\mathbb{E}_{q_{x_s}(x_s)}[\ell(h(x_s), \tilde{f}_t(x_s))], R_s(h, \tilde{f}_t)\}. \tag{17}$$

Then, we give another lemma to show relation between $\tilde{R}_t^{\mathrm{po}}(h, \tilde{f}_t, u_s)$ and $R_t(h, \tilde{f}_t)$.

**Lemma 2.** *Given the multivariate random variable $(X_t, U_t)$ with the probability $\tilde{p}_s^{po}(x_t, u_t)$ and Eq. (15), if $\mathbb{E}_{p'_{x_t}(x_t)}[\ell(h(x_t), f_t(x_t))] \leq R_t(h, f_t) + \rho_{01}^s M_t$, then we have*

$$|\tilde{R}_t^{po}(h, \tilde{f}_t, u_t) - R_t(h, f_t)| \leq \rho_{01}^t \max\{\mathbb{E}_{q_{x_t}(x_t)}[\ell(h(x_t), \tilde{f}_t(x_t))], R_t(h, f_t)\} + \rho_{11}^t \rho_{01}^s M_t. \qquad (18)$$

*Proof.* According to definition of $\tilde{R}_t^{po}(h, \tilde{f}_t, u_t)$ in Theorem 3, we have

$$\tilde{R}_t^{po}(h, \tilde{f}_t, u_t) = (1 - \rho_{u_t})^{-1} \int_{\mathcal{X}} \sum_{u_t=0}^{1} u_t \ell(h(x_t), \tilde{f}_t(x_t)) \tilde{p}_t^{po}(x_t, u_t) dx_t$$

$$= (1 - \rho_{u_t})^{-1} \int_{\mathcal{X}} \ell(h(x_t), \tilde{f}_t(x_t)) \tilde{p}_{X_t|U_t}^{po}(x_t|1) \tilde{p}_{U_t}^{po}(1) dx_t$$

$$\overset{(a)}{=} (1 - \rho_{u_t})^{-1} (1 - \rho_{u_t}) \int_{\mathcal{X}} \ell(h(x_s), \tilde{f}_t(x_t)) \big( \rho_{01}^t q_{x_t}(x_t) + \rho_{11}^t p_{X_t|V_t}^{po}(x_t|1) \big) dx_t$$

$$= \rho_{01}^t \mathbb{E}_{q_{x_t}(x_t)}[\ell(h(x_t), \tilde{f}_t(x_t))] + \rho_{11}^t \int_{\mathcal{X}} \ell(h(x_t), \tilde{f}_t(x_t)) p_{X_t|V_t}^{po}(x_t|V_t = 1) dx_t$$

$$\overset{(b)}{=} \rho_{01}^t \mathbb{E}_{q_{x_t}(x_t)}[\ell(h(x_t), \tilde{f}_t(x_t))] + \rho_{11}^t \int_{\mathcal{X}} \ell(h(x_t), f_t(x_t)) p_{X_t|V_t}^{po}(x_t|V_t = 1) dx_t$$

$$= \rho_{01}^t \mathbb{E}_{q_{x_t}(x_t)}[\ell(h(x_t), \tilde{f}_t(x_t))] + \rho_{11}^t \int_{\mathcal{X}} \ell(h(x_t), f_t(x_t)) p'_{x_t}(x_t) dx_t$$

$$= \rho_{01}^t \mathbb{E}_{q_{x_t}(x_t)}[\ell(h(x_t), \tilde{f}_t(x_t))] + \rho_{11}^t \mathbb{E}_{p'_{x_t}(x_t)}[\ell(h(x_t), f_t(x_t))], \qquad (19)$$

where $(a)$ is based on the definition of $\rho_{u_s}$ and Eq. (14) and $(b)$ is based on the definition of $V_t$ ($f_t(x_t) = \tilde{f}_t(x_t)$ when $V_t = 1$). Since $\mathbb{E}_{p'_{x_t}(x_t)}[\ell(h(x_t), f_t(x_t))] \leq R_t(h, f_t) + \rho_{01}^s M_t$, we have

$$\tilde{R}_t^{po}(h, \tilde{f}_t, u_t) \leq \rho_{01}^t \mathbb{E}_{q_{x_t}(x_t)}[\ell(h(x_t), \tilde{f}_t(x_t))] + \rho_{11}^t (R_t(h, f_t) + \rho_{01}^s M_t). \qquad (20)$$

Thus, we have

$$\begin{aligned}
|\tilde{R}_t^{po}(h, \tilde{f}_t, u_t) - R_t(h, f_t)| &= |\rho_{01}^t \mathbb{E}_{q_{x_t}(x_t)}[\ell(h(x_t), \tilde{f}_t(x_t))] + \rho_{11}^t \mathbb{E}_{p'_{x_t}(x_t)}[\ell(h(x_t), f_t(x_t))] \\
&\quad - R_t(h, f_t)| \\
&\leq |\rho_{01}^t \mathbb{E}_{q_{x_t}(x_t)}[\ell(h(x_t), \tilde{f}_t(x_t))] + \rho_{11}^t (R_t(h, f_t) + \rho_{01}^s M_t) \\
&\quad - R_t(h, f_t)| \\
&= |\rho_{01}^t (\mathbb{E}_{q_{x_t}(x_t)}[\ell(h(x_t), \tilde{f}_t(x_t))] - R_t(h, f_t)) + \rho_{11}^t \rho_{01}^s M_t| \\
&\leq \rho_{01}^t \max\{\mathbb{E}_{q_{x_t}(x_t)}[\ell(h(x_t), \tilde{f}_t(x_t))], R_t(h, f_t)\} + \rho_{11}^t \rho_{01}^s M_t.
\end{aligned}$$

This lemma is proved. $\qquad\qquad\square$

**Remark 5.** In Lemma 2, the assumption $\mathbb{E}_{p'_{x_t}(x_t)}[\ell(h(x_t), f_t(x_t))] \leq R_t(h, f_t) + \rho_{01}^s M_t$ means that the expect risk restricted in $B$ (i.e., $\mathbb{E}_{p'_{x_t}(x_t)}[\ell(h(x_t), f_t(x_t))]$) can represent the true risk $R_t(h, f_t)$ when $\rho_{01}^s$ is small, where $B = \{x : \tilde{f}_t(x) = f_t(x)\}$. In Butterfly, it is equivalent to that pseudo labels provided by noisy source data are more useful if we can select more correct data from noisy source data. If this assumption fails, we cannot gain useful knowledge from $\tilde{f}_t$ even when we can perfectly select correct data from pseudo-labeled target data ($\rho_{01}^t = 0$).

Inequalities (16), (17) and (18) show that if we can avoid to annotate incorrect data as "correct" ($\rho_{01}^s = 0$ and $\rho_{01}^t = 0$), we have $\tilde{R}_s^{\text{po}}(h, u_s) = R_s(h)$, $\tilde{R}_s^{\text{po}}(h, \tilde{f}_t, u_t) = R_s(h, \tilde{f}_t)$ and $\tilde{R}_t^{\text{po}}(h, \tilde{f}_t, u_t) = R_t(h, f_t)$. Nonetheless, $\rho_{01}^s$ and $\rho_{01}^t$ never equal 0, and $\mathbb{E}_{q_s(x_s, y_s)}[\ell(h(x), y)]$, $\mathbb{E}_{q_{x_s}(x_s)}[\ell(h(x_s), \tilde{f}_t(x_s))]$ and $\mathbb{E}_{q_{x_t}(x_t)}[\ell(h(x_t), \tilde{f}_t(x_t))]$ may equal $+\infty$ for some $h$. In next section, we prove that, under the assumption in Remarks 2 and 3, $\tilde{R}_s^{\text{po}}(h, u_s) \to R_s(h)$, $\tilde{R}_s^{\text{po}}(h, \tilde{f}_t, u_t) \to R_s(h, \tilde{f}_t)$ and $\tilde{R}_t^{\text{po}}(h, \tilde{f}_t, u_t) \to R_t(h, f_t)$ if $\rho_{01}^s \to 0$ and $\rho_{01}^t \to 0$. Moreover, we give a new upper bound of $R_t(h, f_t)$.

### H.3.2 Proof of Theorem 3

Now, we prove Theorem 3 as follows.

*Proof.* We first prove upper bounds of $|\tilde{R}_s^{\text{po}}(h, u_s) - R_s(h)|$, $|\tilde{R}_s^{\text{po}}(h, \tilde{f}_t, u_t) - R_s(h, \tilde{f}_t)|$ and $|\tilde{R}_t^{\text{po}}(h, \tilde{f}_t, u_t) - R_t(h, f_t)|$ under assumptions in Theorem 3.

Based on Lemma 1, we have

$$
\begin{aligned}
|\tilde{R}_s^{\text{po}}(h, u_s) - R_s(h)| &= |\rho_{01}^s \mathbb{E}_{q_s(x_s, y_s)}[\ell(h(x_s), y_s)] - (1 - \rho_{11}^s) R_s(h)| \\
&\leq |\rho_{01}^s (R_s(h) + M_s) - \rho_{01}^s R_s(h)| \\
&= \rho_{01}^s M_s.
\end{aligned}
$$

Similar, we have

$$
|\tilde{R}_s^{\text{po}}(h, \tilde{f}_t, u_t) - R_s(h, \tilde{f}_t)| \leq \rho_{01}^t M_t,
$$

$$
|\tilde{R}_t^{\text{po}}(h, \tilde{f}_t, u_t) - R_t(h, f_t)| \leq \rho_{01}^t M_t + \rho_{11}^t \rho_{01}^s M_t.
$$

Since $M_s$ and $M_t$ are positive constants, it is clear that $\tilde{R}_s^{\text{po}}(h, u_s) \to R_s(h)$, $\tilde{R}_s^{\text{po}}(h, \tilde{f}_t, u_t) \to R_s(h, \tilde{f}_t)$ and $\tilde{R}_t^{\text{po}}(h, \tilde{f}_t, u_t) \to R_t(h, f_t)$ when $\rho_{01}^s \to 0$ and $\rho_{01}^t \to 0$.

Specifically, $\forall \epsilon \in (0, 1)$, let $\delta_t = \epsilon / M_t$ and $\delta_s = \epsilon / \max\{M_s, \rho_{11}^t M_t\}$. When $\rho_{01}^s < \delta_s$ and $\rho_{01}^t < \delta_t$, we have

$$
|\tilde{R}_s^{\text{po}}(h, u_s) - R_s(h)| + |\tilde{R}_s^{\text{po}}(h, \tilde{f}_t, u_t) - R_s(h, \tilde{f}_t)| < 2\epsilon
$$

$$
|\tilde{R}_t^{\text{po}}(h, \tilde{f}_t, u_t) - R_t(h, f_t)| < 2\epsilon.
$$

Hence, we prove the Eq. (8). In following, we give a new upper bound of $R_t(h, f_t)$. Call back to Theorem 2, we replace 1) $\tilde{R}_s(h)$ with $\tilde{R}_s^{\text{po}}(h, u_s)$, 2) $\tilde{R}_s(h, \tilde{f}_t)$ with $\tilde{R}_s^{\text{po}}(h, \tilde{f}_t, u_t)$, 3) $R_s(h, \tilde{f}_t)$ with $\tilde{R}_t^{\text{po}}(h, \tilde{f}_t, u_t)$. Then, we have

$$
\begin{aligned}
R_t(h, f_t) \leq{}& \tilde{R}_s^{\text{po}}(h, u_s) + |\tilde{R}_t^{\text{po}}(h, \tilde{f}_t, u_t) - \tilde{R}_s^{\text{po}}(h, \tilde{f}_t, u_t))| + |R_s(h, \tilde{f}_t) - R_s(h)| \\
&+ |\tilde{R}_s^{\text{po}}(h, u_s) - R_s(h)| + |\tilde{R}_s^{\text{po}}(h, \tilde{f}_t, u_t) - R_s(h, \tilde{f}_t)| + |R_t(h, f_t) - \tilde{R}_t^{\text{po}}(h, \tilde{f}_t, u_t)|.
\end{aligned}
$$

Let $\rho_{01}^s \leq \delta_s$ and $\rho_{01}^t \leq \delta_t$, we have

$$
R_t(h, f_t) \leq \underbrace{\tilde{R}_s^{\text{po}}(h, u_s)}_{(i) \text{ risk on noisy data}} + \underbrace{|\tilde{R}_t^{\text{po}}(h, \tilde{f}_t, u_t) - \tilde{R}_s^{\text{po}}(h, \tilde{f}_t, u_s)|}_{(ii) \text{ discrepancy between distributions}} + \underbrace{|R_s(h, \tilde{f}_t) - R_s(h)|}_{(iii) \text{ domain dissimilarity}}
$$

$$
+ \underbrace{2\epsilon}_{(iv) \text{ noise effects from source } \Delta_s} + \underbrace{2\epsilon}_{(v) \text{ noise effects from target } \Delta_t},
$$

Hence, we prove this theorem. $\qquad\square$

