# OpenReview forum: "Wildly Unsupervised Domain Adaptation and Its Powerful and Efficient Solution"
_ICLR.cc/2020/Conference — Reject_

### Official Review · AnonReviewer2 · 2019-10-17
**Official Blind Review #2**

**Rating:** 1

**Review:**

This paper introduces the idea of wildly unsupervised domain adaptation, where the source labels are noisy and the target data is unsupervised. To tackle this, the authors propose an architecture based one two branches: one acting on the mixed source-target data and the other on the target data only. During training, each branch is updated using the idea of co-teaching, by finding the samples with the lowest loss values. Pseudo-labeling is then applied to the target data, and the process iterates.

Originality:
- In essence, the proposed method combines co-teaching (Han et al., 2018) and pseudo-labeling (Saito et al., 2017). While it goes beyond the naive two-stage approach, used here as a baseline, the technical novelty remains limited.
- The main novelty consists of using two branches to model the two domains. However, the necessity for the second branch is not very clearly explained, and remains obscure to me, since the first branch already acts on the mixed source-target data.

Clarity:
In addition the fact that, as mentioned above, the design of the overall framework is not entirely well motivated, I found the paper somewhat hard to follow. In particular:
- One has to go to the appendix to find Alg. 2, which describes one of the key components (the Checking method in Alg. 1). The steps performed by Alg. 2 are not explained anywhere.
- In Alg. 1, it seems surprising that \tilde{D}_T^l is initialized as \tilde{D}_s, since, according to Fig. 3, the second branch should act on the target data only.
- In Alg. 1, the authors do not explain how they initialize the values R(T) and R_t(T).
- I would expect that, to obtain meaningful results from the Checking method, the parameters of the networks F_1, F_2, F_{t1} and F_{t2} should already be initialized to reasonable values. Can the authors comment on the initialization procedure?
- In Alg. 2, how are the inner minimization problems solved? Are u_1 and u_2 truly enforced to be binary variables? How fast can one obtain the solutions to these problems?
- In Fig. 1, it is not clear to me what the term Interaction between the two branches represent. From the text, I could not find any reference to explicit interactions between the branches.
- In Eq. 1, the loss function \ell() is not defined (although I imagine that it is the cross-entropy).

Experiments:
- The experiments show the good behavior of the method. However, while I understand the motivation behind defining the two-stage baseline using ATDA, which is used in the proposed method, there seem to be no strict constrain on using this specific method in the two-stage scenario. For example, based on the results in Table 1, one might rather want to use TCL as the second stage, i.e., have a baseline Co+TCL.
- As I mentioned before, the motivation behind the second branch in the framework is not clear to me. I would appreciate it if the authors could explain the reasoning behind this branch and evaluate their method without it.

Summary:
The novelty of the proposed method, relying on a combination of co-teaching and pseudo-labeling, is limited. Furthermore, the clarity of the paper could be significantly improved.


**Experience Assessment:**

I have published in this field for several years.

**Review Assessment: Checking Correctness Of Derivations And Theory:**

I carefully checked the derivations and theory.

**Review Assessment: Checking Correctness Of Experiments:**

I assessed the sensibility of the experiments.

**Review Assessment: Thoroughness In Paper Reading:**

I read the paper thoroughly.

---

> ### Author Response · Authors · 2019-11-10
> **Response to clarity**
>
> We thank R2 for listing some unclear points in our paper. We will clarify these points here and update them in the revision.
>
> Question 1: In Alg. 1, the authors do not explain how they initialize the values R(T) and R_t(T).
> Answer 1: R(T) and R_t(T) are two functions with respective to T. We do not need to initialize them. We have demonstrated how to select hyper-parameters of R(T) and R_t(T) in Section G.2. Please see page 20 in the revision.
>
> Question 2: I would expect that, to obtain meaningful results from the Checking method, the parameters of the networks F_1, F_2, F_{t1} and F_{t2} should already be initialized to reasonable values. Can the authors comment on the initialization procedure?
> Answer 2: We do not use some special ways to initialize parameters of four networks. All parameters are initialized by the default procedures in TensorFlow on Python 3.6.
>
> Question 3: In Alg. 2, how are the inner minimization problems solved? Are u_1 and u_2 truly enforced to be binary variables? How fast can one obtain the solutions to these problems?
> Answer 3: u_1 and u_2 are binary variables and we can easily solve this minimization problem via using a sorting algorithm.
> Specifically, in \mathcal{L}, we have n instances: (x_i, y_i), where i = 1, 2, …, n. For the ith instance, we will compute its cross-entropy loss (i.e., \ell(F(x_i),y_i)), and we will denote this instance as "selected" if u_i = 1. So, the nature of \mathcal{L} is actually the average value of cross-entropy losses of "selected" instances. In lines 2-3 in Alg. 2, we show that we can obtain u_i by solving a minimization problem: $\textbf{u} = \arg\min_{\textbf{u}':\textbf{1}\textbf{u}'>\alpha|D|}\mathcal{L}(\theta,\textbf{u}';F, D)$, where $\textbf{1}$ is a 1-by-n vector whose elements are 1, $\textbf{u}=[$u_1,...,u_n$]^T$, n=|D| and $\textbf{1}\textbf{u}'$ is the number of 1 in vector $\textbf{u}'$. Recall the nature of the loss \mathcal{L}, we know \mathcal{L} is the average value of cross-entropy losses of "selected" instances, and  $\textbf{1}\textbf{u}'$ is the number of these "selected" instances. Thus this minimization problem is equivalent to "given a fixed network F (F_1 or F_2) and n instances in D, how to select at least k instances such that \mathcal{L} is minimized", where k = $\lceil \alpha|D| \rceil$. To solve this problem, we first use a sorting algorithm (top_k function in TensorFlow) to sort these n instances according to their cross-entropy losses \ell(F_1(x_i),y_i). Then, we select k instance with the smallest cross-entropy losses. Finally, let u_i of these k instances be 1 and u_i of the other instances be 0, and we can get the best $\textbf{u}=[$u_1,...,u_n$]^T$. The average value of cross-entropy losses of these k instances is the minimized value of $\mathcal{L}(\theta,\textbf{u}^{\prime};F, D)$ under the constrain $\textbf{1}\textbf{u}^{\prime}>\alpha|D|$.
> Since we only need to sort |$D$| numbers, we can solve this minimization problem quickly.
> Please see "Solution to minimization problems in Algorithm 2" at page 5 in the revision.
>
> Question 4: One has to go to the appendix to find Alg. 2, which describes one of the key components (the Checking method in Alg. 1). The steps performed by Alg. 2 are not explained anywhere.
> Answer 4: Alg. 2 has been moved back to main context (see page 6 in the revision).  We also explained each line in Alg. 2 in "checking process in Butterfly" at page 5 in the revision.
>
> Question 5: In Fig. 1, it is not clear to me what the term Interaction between the two branches represent. From the text, I could not find any reference to explicit interactions between the branches. In Alg. 1, it seems surprising that \tilde{D}_T^l is initialized as \tilde{D}_s, since, according to Fig. 3, the second branch should act on the target data only.
> Answer 5: The interaction between DIR and TSR happens via the shared CNN. Since we do not have any pseudo-labeled target data, we need to use source data as the pseudo-labeled target data in the first step (i.e. T=1), which follows the ATDA method. We have explain this in Fig. 3 (please see page 4 in the revision) and the description of Alg. 1 in the revision (please see the third paragraph at page 5 in the revision).
>
> Question 6: In Eq. 1, the loss function \ell() is not defined (although I imagine that it is the cross-entropy).
> Answer 6: \ell() is indeed the cross-entropy loss. We have explained this at page 4 in the revision.

---

> ### Author Response · Authors · 2019-11-10
> **Response to experiments**
>
> R2 suggests a baseline and wish to see what will happen if we turn off the branch II. We are running our method on the baseline. We actually have justified what will happen if we turn off the branch II in our ablation study.
>
> Question 1: The experiments show the good behavior of the method. However, while I understand the motivation behind defining the two-stage baseline using ATDA, which is used in the proposed method, there seem to be no strict constrain on using this specific method in the two-stage scenario. For example, based on the results in Table 1, one might rather want to use TCL as the second stage, i.e., have a baseline Co+TCL.
> Answer 1: It is a good suggestion for us. We have added Co+TCL as a baseline (see "baselines" at page 7 in the revision). We also updated the results in the revision.
> Please see Table 1 at page 8, Table 2 at page 9, Tables 5 and 6 at page 18 in the revision.
>
> Question 2: As I mentioned before, the motivation behind the second branch in the framework is not clear to me. I would appreciate it if the authors could explain the reasoning behind this branch and evaluate their method without it.
> Answer 2: we have already justified the necessity of Branch II in our ablation study in Table 4, page 9. In Table 4, B-Net-M represents the situation that we only check the mixed source-target data. From the reported results, Branch II is meaningful and can help improve the target-domain accuracy.
> If we only consider the first branch, we can only obtain the high-quality domain-invariant representation. However, if we consider two branches, we can obtain not only the high-quality domain-invariant representation (DIR) but also the high-quality target-specific representation (TSR). Since we consider to accurately annotate the target data, it is necessary to obtain the high-quality TSR, which is also verified in ATDA method.

---

> ### Author Response · Authors · 2019-11-10
> **Response to "necessity of Branch II"**
>
> One major concern of R2 is that he/she cannot agree with us about the necessity of Branch II of our method since the first branch already acts on the mixed source-target data.
> However, we have already justified the necessity of Branch II in our ablation study in Table 4, page 9. In Table 4, B-Net-M represents the method that we only check the mixed source-target data. From the reported results, Branch II is meaningful and can help improve the target-domain accuracy.
> If we only consider the first branch, we can only obtain the high-quality domain-invariant representation. However, if we consider two branches, we can obtain not only the high-quality domain-invariant representation (DIR) but also the high-quality target-specific representation (TSR). Since we consider to accurately annotate the target data, it is necessary to obtain the high-quality TSR, which is also verified in ATDA method.

---

### Official Review · AnonReviewer1 · 2019-10-21
**Official Blind Review #1**

**Rating:** 8

**Review:**

This paper proposes a new problem setting in the domain adaptation field. Since it is impossible to obtain perfectly clean labeled source data in the real world, existing UDA methods cannot well handle real-world data. However, in wildly unsupervised domain adaptation (WUDA), we do not need a perfectly clean source data, which means that WUDA problem is a more general and realistic problem than existing ones.

To address WUDA problem, the authors proposed Butterfly framework (based on dual checking principle) to simultaneously reduce the 1) noise effects in source domain and 2) distributional discrepancy between source and target domains. They tested the proposed method on simulated and real-world WUDA tasks (35 tasks in total), and the accuracy of proposed method is higher than those of representative UDA methods. They claim that Butterfly can eliminate noise effect, which is strongly supported by Figures 2, 4 and 5. They also present the ablation study to show that each part in Butterfly is meaningful.

In general, this paper is easy to follow and clearly presents the main idea and learning procedures of Butterfly. Since WUDA, as a new problem, could lead a new research direction in the domain adaptation field, this paper should be presented in ICLR 2020. Detailed comments can be seen below.

Pros:

+ WUDA, as a new problem, is very important for the domain adaptation field.
+ Butterfly, as a solution to WUDA, outperforms representative UDA methods on simulated and real-world WUDA tasks.
+ All claims are strongly supported by experimental results, and ablation study shows that each part in Butterfly is a necessary component.
+ Following Ben-David's paper, this paper also presents an upper bound of the target domain risk. This is 1) the first WUDA bound and 2) probably the first DA bound related to pseudo labelling function. The conditions in Remark 3 are very interesting.
+ It is very nice to use noise effect \Delta to explain the abnormal phenomenon in experiments.

Cons:

- The color scheme of figures is not friendly to color-blind people. The authors should do different line styles or marker styles.
- The interaction between DIR and TSR happens via shared CNN, right? The authors should explain this in the caption of Figure 3.
- When T = 1, you directly use source data as the pseudo-labeled target data since there are no pseudo-labeled target data in the first step (based on Algorithm 1). Is that correct? If yes, the authors should explain this in Figure 3 and the description of Algorithm 1. If no, please give a detailed explanation.


**Experience Assessment:**

I have published one or two papers in this area.

**Review Assessment: Checking Correctness Of Derivations And Theory:**

I carefully checked the derivations and theory.

**Review Assessment: Checking Correctness Of Experiments:**

I carefully checked the experiments.

**Review Assessment: Thoroughness In Paper Reading:**

I read the paper thoroughly.

---

> ### Author Response · Authors · 2019-11-10
> **Response to R1**
>
> We appreciate that R1 strongly supports our paper and argue that WUDA, as a new problem, could lead a new research direction in the domain adaptation field. We will address your major concerns here and update the revision.
>
> Question 1: The interaction between DIR and TSR happens via shared CNN, right? The authors should explain this in the caption of Figure 3.
> Answer 1: Yes, your understanding is correct. The interaction between DIR and TSR happens via shared CNN. We have explained this in the caption of Figure 3 in the revision.
> Please see Figure 3, page 4, in the revision.
>
> Question 2: When T = 1, you directly use source data as the pseudo-labeled target data since there are no pseudo-labeled target data in the first step (based on Algorithm 1). Is that correct? If yes, the authors should explain this in Figure 3 and the description of Algorithm 1. If no, please give a detailed explanation.
> Answer 2: Yes, your understanding is correct. Since we do not have any pseudo-labeled target data, we need to use source data as the pseudo-labeled target data in the first step (i.e. T=1). We have explained this in Figure 3 and the description of Algorithm 1 in the revision.
> Please see Figure 3 at page 4, and third paragraph at page 5 in the revision.

---

### Official Review · AnonReviewer3 · 2019-10-23
**Official Blind Review #3**

**Rating:** 3

**Review:**

This paper presents a method for unsupervised domain adaptation. The problem is well known in literature and follows the setting of labeled source and unlabeled target set. This work proposes the “butterfly network” suitable to train on noisy data (labels) and assign pseudo-labels to the target set. The butterfly network consists in two branches one for source+target and one for target only data. Both use the same optimization objective and a “checking” mechanism has been devised for pseudo-labelling the data.
Decision: weak reject
Motivation: This paper has some merit but does not present the method in a clear way, it requires some additional effort to go through the method and retrieve from the appendix information useful for full understanding. For example algorithm 2 is a key component of the method and is in appendix, notation is not always clear nor explained (the loss in algorithm 2 is ), the networks F_1, F_2, F_t1, F_t2 could be added in figure 3 for more clear understanding and R(t) along with u_i could be more clearly defined (how to obtain u_i is still not clear to me after several re-reading of the paper). The introduced losses should be better justified.
Anyway, as I said the paper has some merit, it provides many insights and extensive analysis on “butterfly” method for unsupervised domain adaptation. The experimental section is extensive and demonstrates improvement in performance using this method compared to state-of-the-art, even though for some "real world" datasets (e.g. SUN, Caltech, ImageNet) the improvement is not so significant as in the case of MNIST-SYND. Could the authors provide results on MNIST-SVHN to help compare with other papers in literature? Did the authors observe the same evolution of the accuracy (decreasing and increasing after a few epochs) also on the "real world" datasets as in MNIST-SYND?
Replicability: as I said the method is not really clearly explained and therefore I am not confident I could implement and replicate the results. This not because I think the method is complex but because some key components that I pointed out previously about the method are not clear and I strogly believe these are key components to replicate the results.


**Experience Assessment:**

I have published in this field for several years.

**Review Assessment: Checking Correctness Of Derivations And Theory:**

I assessed the sensibility of the derivations and theory.

**Review Assessment: Checking Correctness Of Experiments:**

I assessed the sensibility of the experiments.

**Review Assessment: Thoroughness In Paper Reading:**

I read the paper thoroughly.

---

> ### Author Response · Authors · 2019-11-10
> **Response to experiments**
>
> The other concerns of R3 are related to experiments. We will address these concerns here and update the revision.
> Question 1: Did the authors observe the same evolution of the accuracy (decreasing and increasing after a few epochs) also on the "real world" datasets as in MNIST-SYND?
> Answer 1: Yes, we also observe this phenomenon on the real world datasets.
>
> Question 2:  Could the authors provide results on MNIST-SVHN to help compare with other papers in literature?
> Answer 2: We have added this experiment.
> Please see the target-domain accuracy on both UDA tasks in Table 3, page 9, in the revision. We also describe this table at page 8 in the revision ("Results on two UDA tasks").

---

> ### Author Response · Authors · 2019-11-10
> **Response to unclear descriptions**
>
> The major concern of R3 is that he/she thinks we do not clearly describe 1) the introduced loss, 2) how to obtain u_i, 3) the underlying meaning of R(t) and 4) notations in Algorithm 2. We address these concerns here and update these new descriptions in the revision.
>
> Question 1: The introduced losses should be better justified (the nature of loss).
> Answer 1: In loss function \mathcal{L}, we have n instances: (x_i, y_i), where i = 1, 2, …, n. For the ith instance, we will compute its cross-entropy loss (i.e., \ell(F(x_i),y_i)), and we will denote this instance as "selected" if u_i = 1. Thus, the nature of \mathcal{L} is actually the average value of cross-entropy loss of these "selected" instances.
> Please see "nature of the loss \mathcal{L}" at the end of page 4 in the revision.
>
> Question 2: How to obtain u_i is still not clear to me after several re-reading of the paper.
> Answer 2: In lines 2-3 in Algorithm 2, we show that we can obtain u_i by solving a minimization problem: $\textbf{u} = \arg\min_{\textbf{u}^{\prime}:\textbf{1}\textbf{u}^{\prime}>\alpha|D|}\mathcal{L}(\theta,\textbf{u}^{\prime};F, D)$, where $\textbf{1}$ is a 1-by-n vector whose elements are 1, $\textbf{u}=[$u_1,...,u_n$]^T$, n=|D| and $\textbf{1}\textbf{u}^{\prime}$ represents the number of 1 in the vector $\textbf{u}^{\prime}$. Recall the nature of the loss \mathcal{L} (Answer 1), we know \mathcal{L} is the average value of cross-entropy losses of "selected" instances, and  $\textbf{1}\textbf{u}'$ is the number of these "selected" instances. Thus this minimization problem is equivalent to "given a fixed network F (F_1 or F_2) and n instances in D, how to select at least k instances such that \mathcal{L} is minimized", where k = $\lceil \alpha|D| \rceil$. To solve this problem, we first use a sorting algorithm (top_k function in TensorFlow) to sort these n instances according to their cross-entropy losses \ell(F_1(x_i),y_i). Then, we select k instance with the smallest cross-entropy losses. Finally, let u_i of these k instances be 1 and u_i of the other instances be 0, and we can get the best $\textbf{u}=[$u_1,...,u_n$]^T$. The average value of cross-entropy losses of these k instances is the minimized value of $\mathcal{L}(\theta,\textbf{u}^{\prime};F, D)$ under the constrain $\textbf{1}\textbf{u}^{\prime}>\alpha|D|$.
> Please see "Solution to minimization problems in Algorithm 2" at page 5 in the revision.
>
> Question 3: R(T) could be more clearly defined.
> Answer 3: R(T) is defined in line 11 of Algorithm 1 in page 5. We will explain it in detail here. In general, R(T) is a piecewise-defined linear function. Specifically, when T>=T_k, R(T) = 1-\tau; when T<T_k, R(T) = 1 - T/T_k * \tau.
> Please see the end of the fourth paragraph at page 4 in the revision.
>
> Question 4: Algorithm 2 is a key component of the method and is in appendix, notation is not always clear nor explained.
> Answer 4: Many thanks for your suggestions. We have moved Algorithm 2 to the main context (please see page 6 in the revision). We also explained each line in Algorithm 2 in "checking process in Butterfly" at page 5 in the revision.

---

### Author Response · Authors · 2019-11-09
**Revision of figures**

According to valuable comments from R1 and R3,
1) we have added the networks F_1, F_2, F_t1, F_t2 in Figure 3 for more clear understanding (see Figure 3, page 4, in the revision);
2) we have explained interaction between DIR and TSR in the caption of figure 3 (see Figure 3, page 4, in the revision);
3) we have explained that, when T = 1, we directly use source data as the pseudo-labeled target data in the caption of figure 3 (see Figure 3, page 4, in the revision);
4) we have changed color scheme of figures such that they are friendly to color-blind people (see Figure 2, page 3, and Figure 4, page 7, and Figure 5, page 8, in the revision).

---

### Author Response · Authors · 2019-11-10
**Thank reviewers for their valuable comments**

We first thank all three reviewers for their valuable comments. We have tried our best to address your concerns. Please see our responses below. In the revision, we highlighted the revised content by changing the color of font to blue. Please check our revision.

---

### Author Response · Authors · 2019-11-15
**The revision has been uploaded**

According to valuable comments from three reviewers, we have revised our paper. In this revision, we demonstrate more details related to our method, which should help reviewers better understand our proposal. We also revise our previous responses according to the revision. Please have a look.

---

### Decision · Program_Chairs · 2019-12-19

**Decision:**

Reject

**Comment:**

The authors proposed a new problem setting called Wildly UDA (WUDA) where the labels in the source domain are noisy. They then proposed the "butterfly" method, combining co-teaching with pseudo labeling and evaluated the method on a range of WUDA problem setup. In general, there is a concern that Butterfly as the combination between co-teaching and pseudo labeling is weak on the novelty side. In this case the value of the method can be assessed by strong empirical result.  However as pointed out by Reviewer 3, a common setup (SVHN<-> MNIST) that appeared in many UDA paper was missing in the original draft. The author added the result for SVHN<-> MNIST  as  a response to review 3, however they only considered the UDA setting, not WUDA, hence the value of that experiment was limited. In addition, there are other UDA methods that achieve significantly better performance on SVHN<->MNIST that should be considered among the baselines. For example DIRT-T (Shu et al 2018) has a second phase where the decision boundary on the target domain is adjusted, and that could provide some robustness against a decision boundary affected by noise.

Shu et al (2018) A DIRT-T Approach to Unsupervised Domain Adaptation. ICLR 2018. https://arxiv.org/abs/1802.08735

I suggest that the authors consider performing the full experiment with WUDA using SVHN<->MNIST, and also consider the use of stronger UDA methods among the baseline.